# Deep Learning and Quantum Entanglement: Fundamental Connections with Implications to Network Design

**Yoav Levine, David Yakira, Nadav Cohen & Amnon Shashua**
The Hebrew University of Jerusalem
`{yoavlevine,davidyakira,cohennadav,shashua}@cs.huji.ac.il`

## Abstract

Formal understanding of the inductive bias behind deep convolutional networks, i.e. the relation between the network's architectural features and the functions it is able to model, is limited. In this work, we establish a fundamental connection between the fields of quantum physics and deep learning, and use it for obtaining novel theoretical observations regarding the inductive bias of convolutional networks. Specifically, we show a structural equivalence between the function realized by a convolutional arithmetic circuit (ConvAC) and a quantum many-body wave function, which facilitates the use of quantum entanglement measures as quantifiers of a deep network's expressive ability to model correlations. Furthermore, the construction of a deep ConvAC in terms of a quantum Tensor Network is enabled. This allows us to perform a graph-theoretic analysis of a convolutional network, tying its expressiveness to a min-cut in its underlying graph. We demonstrate a practical outcome in the form of a direct control over the inductive bias via the number of channels (width) of each layer. We empirically validate our findings on standard convolutional networks which involve ReLU activations and max pooling. The description of a deep convolutional network in well-defined graph-theoretic tools and the structural connection to quantum entanglement, are two interdisciplinary bridges that are brought forth by this work.

## 1 Introduction

A central factor in the application of machine learning to a given task is the restriction of the hypothesis space of learned functions known as *inductive bias*. In deep convolutional networks, inductive bias manifests itself in architectural features such as number of layers, number of channels per layer, and more (LeCun et al., 2015). Formal understanding of the inductive bias behind convolutional networks is limited – the assumptions encoded into these models, which seem to form an excellent prior knowledge for different types of data (*e.g.* Krizhevsky et al. (2012); He et al. (2016); van den Oord et al. (2016)), are for the most part a mystery.

An important aspect of the influence that a certain architectural feature has on the inductive bias, is its effect on the network's ability to model correlations between regions of its input. In this regard, one typically considers partitions that divide input regions into disjoint sets, and asks how far the function realized by the network is from being separable with respect to these partitions(Cohen and Shashua, 2017; Levine et al., 2017). For example, Cohen and Shashua (2017) show that when separability is measured through the algebraic notion of separation-rank, deep Convolutional Arithmetic Circuits (ConvACs) (Cohen et al., 2016b) support exponential (in network size) separation-ranks for certain input partitions, while being limited to polynomial separation-ranks for others. ConvACs are a special class of convolutional networks, characterized by linear activations and product pooling, which served a key role in theoretical analyses of convolutional networks, in virtue of their algebraic structure.

In this work, we draw upon formal similarities between how physicists describe a system of many-particles as a quantum mechanical wave function, and how machine learning practitioners map a high-dimensional input (*e.g.* image) to a set of output labels through a deep network. In particular, we show that there is a structural equivalence between a function modeled by a ConvAC and a many-body quantum wave function, which relies on their underlying tensorial structure. This allows employment of the well-established physical notion of quantum entanglement measures (Plenio and Virmani, 2007), which subsumes other algebraic notions of separability such as the separation-rank mentioned above, for the analysis of correlations modeled by deep convolutional networks.

Importantly, quantum entanglement is used by physicists as prior knowledge to form compact representations of many-body wave functions in what is known as *Tensor Networks* (TNs) (Östlund and Rommer, 1995; Verstraete and Cirac, 2004; Vidal, 2008; Evenbly and Vidal, 2011). In the domain of machine learning, a network in the form of a ConvAC is effectively a compact representation of a multi-dimensional array related to the convolutional weights. This has been analyzed to date via tensor decompositions – where the representations are based on linear combinations of *outer-products* between lower-order tensors (Cohen et al., 2016b). A TN, on the other hand, is a way to compactly represent a higher-order tensor through *inner-products* among lower-order tensors, which allows a natural representation of TNs through an underlying graph. Although the fundamental language is different, we show that a ConvAC can be mapped to a TN, and thus a graph-theoretic setting for studying functions modeled by deep convolutional networks is brought forth. In particular, notions of max-flow/min-cut are shown to convey important meaning.

The results we present, connect the inductive bias of deep convolutional networks to the number of channels in each layer, and indicate how these should be set in order to satisfy prior knowledge on the task at hand. Specifically, the ability of a ConvAC to represent correlations between input regions is shown to be related to a min-cut over all edge-cut sets that separate the corresponding input nodes in the associated TN. Such results enable one to avoid bottle-necks and adequately tailor the network architecture through application of prior knowledge. Our results are theoretically proven for a deep ConvAC architecture; their applicability to a conventional deep convolutional network architecture, which involves ReLU activations and max pooling, is demonstrated through experiments.

Some empirical reasoning regarding the influence of the channel numbers on the network's performance has been suggested (e.g. Szegedy et al. (2016)), mainly regarding the issue of bottle-necks which is naturally explained via our theoretical analysis below. Such insights on the architectural design of deep networks are new to machine learning literature, and rely on TN bounds recently derived in physics literature, referred to as 'quantum min-cut max-flow' (Cui et al., 2016). The mapping we present between ConvACs and TNs indicates new possibilities for the use of graph-theory in deep networks, where min-cut analysis could be just the beginning. Additionally, the connections we derive to quantum entanglement and quantum TNs may open the door to further well-established physical insights regarding correlation structures modeled by deep networks.

The use of TNs in machine learning has appeared in an empirical context where Stoudenmire and Schwab (2016) trained a matrix product state (MPS) TN architecture to perform supervised learning tasks on the MNIST data-set. Additionally, there is a growing interest in the physics community in RBM based forms for variational many-body wave functions (*e.g.* Carleo and Troyer (2017)). Chen et al. (2017) present a theoretical mapping between RBMs and TNs which allows them to connect the entanglement bounds of a TN state to the expressiveness of the corresponding RBM.

## 2   PRELIMINARIES

We provide below the minimal tensor analysis background required for following the analyses of ConvACs and TNs that are carried out in this paper. The core concept in tensor analysis is a *tensor*, which may be thought of as a multi-dimensional array. The *order* of a tensor is defined to be the number of indexing entries in the array, which are referred to as *modes*. The *dimension* of a tensor in a particular mode is defined as the number of values that may be taken by the index in that mode. If $\mathcal{A}$ is a tensor of order $N$ and dimension $M_i$ in each mode $i \in [N]$, its entries are denoted $\mathcal{A}_{d_1 \ldots d_N}$, where the index in each mode takes values between 1 and the appropriate dimension, $d_i \in [M_i]$. Suppose $\mathcal{A}$ is a tensor of order $N$, and let $(A, B)$ be a partition of $[N] := \{1, \ldots, N\}$, *i.e.* $A$

$$M\mathbf{v} = \mathbf{u}$$
$$\sum_{k=1}^{r_1} M_{dk} v_k = u_d$$

Figure 1: Contraction of a Simple TN.

and $B$ are disjoint subsets of $[N]$ whose union covers the entire set. The *matricization of $\mathcal{A}$ w.r.t. the partition $(A, B)$*, denoted $[\![\mathcal{A}]\!]_{A,B}$, is essentially the arrangement of the tensor elements as a matrix whose rows correspond to $A$ and columns to $B$ (see appendix A for exact definition).

A TN (see overview in Orús (2014) is a weighted graph, where each node corresponds to a tensor whose order is equal to the degree of the node in the graph. Accordingly, the edges emanating out of a node, also referred to as its legs, represent the different modes of the corresponding tensor. The weight of each edge in the graph, is equal to the dimension of the appropriate tensor mode.

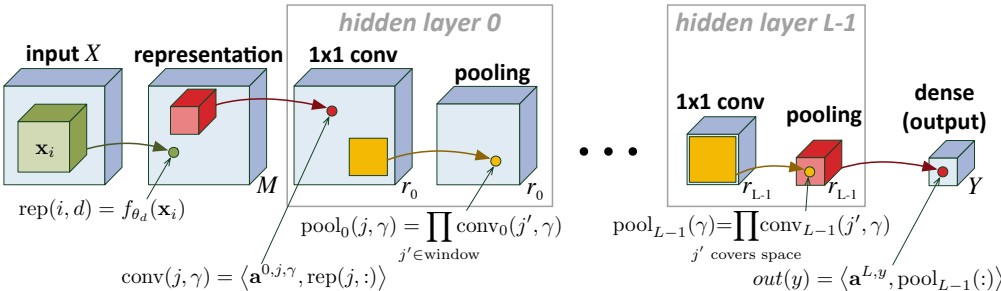

Figure 2: The Convolutional Arithmetic Circuit (ConvAC) network (Cohen et al., 2016b).

Moving on to the connectivity properties of a TN, edges which connect two nodes in the TN represent an operation between the two corresponding tensors. A index which represents such an edge is called a contracted index, and the operation of contracting that index is a summation over all of the values it can take. An index representing an edge with one loose end is called an open index. The tensor represented by the entire TN, whose order is equal to the number of open indices, can be calculated by summing over all of the contracted indices in the network. In fig. 1, a TN corresponding to the operation of multiplying a vector $\mathbf{v} \in \mathbb{R}^{r_1}$ by a matrix $M \in \mathbb{R}^{r_2 \times r_1}$ is depicted. The computation is performed by summing over the only contracted index, $k$. Since there is only one open index, $d$, the result of contracting the network is an order 1 tensor (a vector): $\mathbf{u} \in \mathbb{R}^{r_2}$ which upholds $\mathbf{u} = M\mathbf{v}$. Though we use below the contraction of indices in a more elaborate TN, this operation can be essentially viewed as a generalization of matrix multiplication.

## 3    CONVOLUTIONAL NETWORKS AND QUANTUM WAVE FUNCTIONS

When describing the quantum mechanical properties of a system composed of many interacting particles, referred to as a *many-body* quantum system, physicists are required to employ functions which are able to express an elaborate relation between the different particles. Similarly, machine learning tasks require functions with the ability to express a complex relation between many input elements, e.g. many pixels in an image. In this section, we formulate this analogy.

Our construction will be based on the ConvAC architecture introduced by Cohen et al. (2016b), illustrated in fig. 2. The ConvAC is a deep convolutional network that operates similarly to a regular convolutional network, only with linear activations and product pooling layers (which introduce the non-linearity) instead of the more common non-linear activations (e.g. ReLU) and average/max pooling. ConvACs are closely related to SimNets (Cohen and Shashua, 2014; Cohen et al., 2016a), and their underlying operations lend themselves to mathematical analyses based on measure theory and tensor analysis. From an empirical perspective, ConvACs work well in many practical settings, e.g. for optimal classification with missing data (Sharir et al.), and for compressed networks (Cohen et al., 2016a). Importantly, through the concept of generalized tensor decompositions, a ConvAC can be transformed to a standard convolutional network with ReLU activation and average/max pooling, which laid the foundation for extending its proof methodologies to such ConvNets (Cohen and Shashua, 2016). This deep learning architecture was chosen for our analysis below due to its underlying tensorial structure which resembles the quantum many-body wave function, as will soon be shown.

The input space of the network, denoted by $X = (\mathbf{x}_1, ..., \mathbf{x}_N)$, can be thought of as an image, where each $\mathbf{x}_j$ corresponds to a local patch from that image. The $Y$ network outputs, denoted by $\mathbf{h}_y(\mathbf{x}_1, ..., \mathbf{x}_N)$ for $y \in [Y]$, are shown in Cohen et al. (2016b) to have the following form:

$$\mathbf{h}_y(\mathbf{x}_1, ..., \mathbf{x}_N) = \sum_{d_1, .., d_N=1}^{M} \mathcal{A}^y_{d_1...d_N} \prod_{j=1}^{N} f_{\theta_{d_j}}(\mathbf{x}_j) = \sum_{d_1, .., d_N=1}^{M} \mathcal{A}^y_{d_1...d_N} \mathcal{A}^{(\text{rank-1})}_{d_1...d_N}(\mathbf{x}_1, ..., \mathbf{x}_N), \quad (1)$$

where $\mathcal{A}^y$ and $\mathcal{A}^{(\text{rank-1})}$ are tensors of order $N$ and dimension $M$ in each mode. The entries of the *conv-weights tensor* $\mathcal{A}^y$ are given by polynomials in the network's convolutional weights, $\mathbf{a}^{l,j,\gamma}$ (see fig. 2 and Cohen et al. (2016b)). The entries of $\mathcal{A}^{(\text{rank-1})}$ are given by the application of the $M$ linearly independent representation functions $\{f_{\theta_d}\}_{d=1}^{M}$ on the input patches, which are an initial mapping of the inputs to an $M$-dimensional feature space.

We now turn to a brief presentation of the methods with which physicists describe the quantum mechanical properties of a many-body system (see appendix B for a more detailed introduction).

A state of a system, which is a complete description of a physical system, is given in quantum mechanics as a wave function, denoted by $|\psi\rangle$. We limit our discussion to states which reside in finite dimensional Hilbert spaces, as these are at the heart of our analogy to convolutional networks. We discuss the case of $N$ particles, each corresponding to a local Hilbert space $\mathcal{H}_j$ for $j \in [N]$ such that $\forall j : \dim(\mathcal{H}_j) = M$. Denoting an orthonormal basis of the local Hilbert space by $\{|\psi_d\rangle\}_{d=1}^M$, the many-body wave function $|\psi\rangle \in \mathcal{H} = \otimes_{j=1}^N \mathcal{H}_j$ can be written as:

$$|\psi\rangle = \sum_{d_1 \ldots d_N = 1}^M \mathcal{A}_{d_1 \ldots d_N} |\psi_{d_1}\rangle \otimes \cdots \otimes |\psi_{d_N}\rangle, \tag{2}$$

where $|\psi_{d_1}\rangle \otimes \cdots \otimes |\psi_{d_N}\rangle$ is a basis vector of the $M^N$ dimensional Hilbert space $\mathcal{H}$, and the *coefficients tensor* $\mathcal{A}_{d_1 \ldots d_N}$ is the tensor holding the corresponding coefficients.

We will tie between the function realized by a ConvAC given in eq. 1, and the many-body quantum wave function given in eq. 2. First, we consider a special case of $N$ particles which exhibit no quantum correlations (to be formulated in section 4 below). The state of such a system is called a *product state*, and can be written down as a single tensor product of local states $|\phi_j\rangle \in \mathcal{H}_j$: $|\psi^{\text{ps}}\rangle = |\phi_1\rangle \otimes \cdots \otimes |\phi_N\rangle$. By expanding each local state in the respective basis, $|\phi_j\rangle = \sum_{d_j=1}^M v_{d_j}^{(j)} |\psi_{d_j}\rangle$, the product state assumes a form similar to eq. 2:

$$|\psi^{\text{ps}}\rangle = \sum_{d_1 \ldots d_N = 1}^M \mathcal{A}_{d_1 \ldots d_N}^{\text{ps}} |\psi_{d_1}\rangle \otimes \cdots \otimes |\psi_{d_N}\rangle, \tag{3}$$

with the entries of its coefficients tensor given by: $\mathcal{A}_{d_1 \ldots d_N}^{\text{ps}} = \prod_{j=1}^N v_{d_j}^{(j)}$. If we compose each local state $|\phi_j\rangle$ such that its projection on the local basis vector equals $v_d^{(j)} = f_{\theta_d}(\mathbf{x}_j)$, then the inner product between the many-body quantum state $|\psi\rangle$ and the tailored product state $|\psi^{\text{ps}}\rangle$ is equal to:

$$\langle \psi^{\text{ps}} | \psi \rangle = \sum_{d_1 \ldots d_N = 1}^M \mathcal{A}_{d_1 \ldots d_N} \mathcal{A}_{d_1 \ldots d_N}^{\text{ps}} (\mathbf{x}_1, \ldots, \mathbf{x}_N) = \sum_{d_1 \ldots d_N = 1}^M \mathcal{A}_{d_1 \ldots d_N} \prod_{j=1}^N f_{\theta_{d_j}}(\mathbf{x}_j), \tag{4}$$

reproducing eq. 1 for a single class $y$, as $\mathcal{A}_{d_1 \ldots d_N}^{\text{ps}} = \mathcal{A}_{d_1 \ldots d_N}^{\text{(rank-1)}}$ by construction. This result ties between the function realized by a convolutional network to that which a many-body wave function models. Specifically, the conv-weights tensor is analogous to the coefficients tensor of the many-body wave function, while the input to the convolutional network is analogous to the constructed product state. In the following sections, we will use this analogy to acquire means of analyzing the expressiveness of a convolutional network via the properties of its underlying tensor.

## 4  CORRELATIONS AND MEASURES OF ENTANGLEMENT

The structural connection between the many-body wave function and the function realized by a ConvAC, presented in the previous section, creates an opportunity to employ well-established physical insights and tools for analyzing the inductive bias of convolutional networks. We present in this section the concept of quantum entanglement measures, and use it to motivate and extend previously suggested means for quantifying correlations of a deep convolutional network.

In Cohen and Shashua (2017); Levine et al. (2017), the algebraic notion of *separation-rank* is used as a tool for measuring correlations modeled by a function between two disjoint parts of its input. Let $f(\cdot)$ be a function over $\mathbf{x}_1 \ldots \mathbf{x}_N$, and let $(A, B)$ be a partition of $[N]$. The separation-rank of $f(\cdot)$ w.r.t. $(A, B)$ measures the strength of correlation that $f(\cdot)$ models between input elements corresponding to $A$ ($\{\mathbf{x}_i\}_{i \in A}$) and those corresponding to $B$ ($\{\mathbf{x}_j\}_{j \in B}$). If $f(\cdot)$ is separable w.r.t. $(A, B)$, meaning there exist functions $g(\cdot)$ and $h(\cdot)$ such that $f(\mathbf{x}_1, \ldots, \mathbf{x}_N) = g((\mathbf{x}_i)_{i \in A}) \cdot h((\mathbf{x}_j)_{j \in B})$, then under $f(\cdot)$ there is absolutely no correlation between the inputs of $A$ and those of $B$.[1] In this case, the separation-rank is equal to 1 by definition. In general, the separation rank of $f(\cdot)$ w.r.t. $(A, B)$ is defined to be the minimal number of summands that together give $f(\cdot)$, where each summand is separable w.r.t. $(A, B)$. Higher separation rank indicates larger deviation from separability, *i.e.* stronger interaction (correlation) modeled between sides of the partition.[2] The analysis of separation ranks allows control over the inductive bias when designing a

---

[1] In a statistical setting, where $f(\cdot)$ is a probability density function, separability w.r.t. $(A, B)$ corresponds to statistical independence between inputs from $A$ and $B$.

[2] See Cohen and Shashua (2017) for a formalization of this argument.

deep network architecture – the network can be designed such that characteristic correlations in the input are modeled, *i.e.* partitions that split correlated regions have high separation ranks.

In the physics domain, special attention is given to the inter-particle correlation structure characterizing a many-body wave function, as it has broad implications regarding the physical properties of the examined system. We present below the concept of *quantum entanglement measures* that is widely used by physicists as a quantifier of correlations in a many-body quantum system. Remarkably, this approach for quantifying correlations is very similar to the above presented tool of the separation-rank, which in fact corresponds to a particular quantum entanglement measure.

Consider a partition of $N$ particles labeled by integers $[N]$, which splits it into two disjoint subsystems $A$ and $B$. Let $\mathcal{H}^A$ and $\mathcal{H}^B$ be the Hilbert spaces corresponding to particles in subsystems $A$ and $B$, respectively. In what is referred to as a 'Schmidt decomposition', the many-body quantum wave function in eq. 2 can be written as (see appendix B.1 for derivation):

$$|\psi\rangle = \sum\nolimits_{\alpha=1}^{r} \lambda_\alpha \left|\phi_\alpha^A\right\rangle \otimes \left|\phi_\alpha^B\right\rangle, \tag{5}$$

where $r := \min(\dim(\mathcal{H}^A), \dim(\mathcal{H}^B))$, $\{\lambda_\alpha\}_{\alpha=1}^{r}$ are the singular values of the matricization $[\![\mathcal{A}]\!]_{A,B}$, and $\{\left|\phi_\alpha^A\right\rangle\}_{\alpha=1}^{r}$, $\{\left|\phi_\alpha^B\right\rangle\}_{\alpha=1}^{r}$ are $r$ vectors in the bases of $\mathcal{H}^A$ and $\mathcal{H}^B$, respectively, obtained by a singular value decomposition. Eq. 5 represents the $N$ particle wave function in terms of a sum of tensor products between two disjoint parts of it. Each summand in eq. 5 is a separable state w.r.t. the partition $(A, B)$, which is analogous to the separable function in the above discussion. Intuitively, as above, the more correlated two sides of a partition are, the more 'complicated' the function describing their relation should be. Essentially, a measure of entanglement w.r.t. the partition $(A, B)$ is a quantity that represents the difference between the state in question and a state that is separable w.r.t. this partition. There are several different measures, such as the entanglement entropy (Vedral and Plenio, 1998) – the entropy of $[\![\mathcal{A}]\!]_{A,B}$'s singular values,[3] the geometric measure (Shimony, 1995) – the minimal $\mathcal{L}^2$ distance of $|\psi\rangle$ from any separable state, and the Schmidt number (Terhal and Horodecki, 2000) – which is simply the number of $[\![\mathcal{A}]\!]_{A,B}$'s non-zero singular values, or equivalently its rank.

This method for quantifying quantum correlations can now be readily transferred into the machine learning domain. Utilizing the structural analogy that was established in section 3, the measures of entanglement constitute an instrument for quantifying the correlations that a convolutional network can model. Specifically, we've shown the conv-weights tensor to be analogous to the coefficients tensor of the many-body wave function, thus the entanglement measures can be analogously defined using the singular values of a matricization of the conv-weights tensor. Since it was shown by Cohen and Shashua (2017) that the separation-rank is equal to the rank of the matricized conv-weights tensor, it is precisely equal to the Schmidt number. The analogy to physics suggests that correlation measures more sensitive than the separation rank may be borrowed, providing a more sensitive algebraic perspective on the hypothesis space of convolutional networks, which takes into account the relative magnitudes of $[\![\mathcal{A}]\!]_{A,B}$'s non-zero singular values and not merely their number.

Physicists have a rich tool-set for exploiting knowledge regarding quantum entanglement measures for the design of computational representations of quantum wave functions. We are now in a position to borrow such tools, and use them for the design of convolutional networks. In particular, we will establish a relation between the correlations modeled by a ConvAC and the widths of its hidden layers, and make use of these relations for controlling the inductive bias of the network.

## 5 Layer Widths Effect on the Expressiveness of a Deep Network

In the previous section, we have seen that the coefficients or conv-weights tensor $\mathcal{A}_{d_1 \dots d_N}$, which has $M^N$ entries, encapsulates the information regarding the correlations of the many-body quantum wave function or of the function realized by a ConvAC. The curse of dimensionality manifests itself in the exponential dependence on the number of particles or image patches. In a quantum many-body setting, this renders impractical the ability to investigate or even store a wave function of more than a few dozens of interacting particles. A common tool used to tackle this problem in the physics community is a Tensor Network, which allows utilizing prior knowledge regarding correlations when attempting to represent an exponentially complicated wave function with a polynomial

---

[3]$|\psi\rangle$ is conventionally chosen to be normalized such that the singular values uphold $\sum_\alpha |\lambda_\alpha|^2 = 1$. This can be relaxed and the entropy may be defined on the normalized singular values.

amount of resources. In appendix C we provide a thorough introduction to TNs, which were briefly introduced in section 2. In this section, we draw inspiration from the physics approach and present a construction of a ConvAC as a TN. This construction will allow us to demonstrate how adequately tailoring the number of channels in each layer of the deep network can enhance its expressivity by fitting the form of the function realized by it to given correlations of the input. In this we show how the parameters of the ConvAC can be most efficiently distributed given prior knowledge on the nature of the input, which is in fact matching the inductive bias to the task at hand.

Fig. 3 shows the TN which represents a one-dimensional[4] network equivalent to the one shown in fig. 2, with pooling windows of size 2 and $N = 8$ (see appendix D for full details of construction). The round (order 2) nodes in each layer represent matrices holding the convolutional weights of that layer. The triangle nodes correspond to a special tensor that hosts 1's on its super-diagonal and 0's elsewhere, effectively enforcing the same channel pooling attribute of the network. The tensor $\mathcal{A}^y_{d_1...d_N}$ is obtained upon summation over all the indices

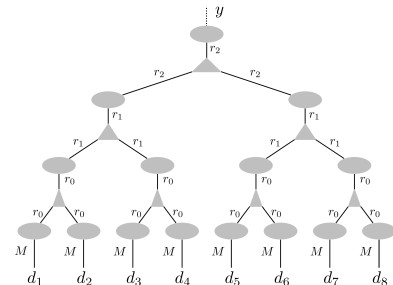

Figure 3: TN of the weights tensor $\mathcal{A}^y_{d_1...d_N}$.

which correspond to internal edges, leaving the external edges which correspond to $d_1, ..., d_N, y$ open. As mentioned above, a TN is a weighted graph, and the weights marked next to each edge in this TN are equal by construction to the number of channels in the corresponding ConvAC layer $l$, denoted $r_l$. This last equivalence will allow us to draw a direct relation between the number of channels in each layer of a deep ConvAC and the functions it is able to model. Accordingly, it will allow us to provide prescriptions regarding the layer widths for the design of a network that is meant to support known input correlations.

Our main result, presented in theorem 1, relies on one of the most recent advances in the study of the quantitative connection between quantum entanglement and TNs, namely 'quantum min-cut max-flow' (Cui et al., 2016). The key accomplishment that the TNs tool brings forth, is the ability to apply graph-theoretic tools to a deep convolutional network. Specifically, we tie the network's ability to model correlations between two disjoint input regions $A$ and $B$, as measured by the Schmidt entanglement measure, to the minimal value, over all cuts separating $A$ from $B$, of the multiplication of the cut edges' weights (the multiplicative minimal cut).

**Theorem 1.** *(proof in appendix E) Let $(A, B)$ be a partition of $[N]$, and $[\![\mathcal{A}^y]\!]_{A,B}$ be the matricization w.r.t. $(A, B)$ of the conv-weights tensor $\mathcal{A}^y$ of the ConvAC depicted in fig. 2 with pooling windows of size 2. Assume that the channel numbers across the layers are all powers of the same integer,[5] and suppose we randomize the network weights by some continuous distribution. Then, with probability 1, the rank of the matricization $[\![\mathcal{A}^y]\!]_{A,B}$ (the Schmidt measure w.r.t. $(A, B)$) is equal to the multiplicative minimal cut separating $A$ from $B$ in the respective TN.*

Theorem 1 leads to practical implications regarding the construction of a deep network architecture when there is prior knowledge on the task at hand. If one wishes to construct a deep ConvAC that is expressive enough to model an intricate correlation structure according to some partition, it is advisable to choose the channel numbers such that the network is able to support such correlations, by ensuring that all the cuts separating these two parts in the corresponding TN have high weights. For example, consider the left-right partition in which $A$ and $B$ hold the left and right input patches, respectively. The multiplicative minimal cut weight is in this case equals $\min(r_{L-1}, r_{L-2}, ..., r_l^{2^{(L-2-l)}}, ..., r_0^{N/4}, M^{N/2})$, where $L := \log_2 N$ (in the example given in fig. 3, $L = 3$). We see that choosing a small number of channels for the deeper layers can create an undesired 'shortcut' which harms the expressiveness of the network in a way that prevents it from modeling the long ranged correlations which correspond to this partition, present for example in symmetric face images. Alternatively, considering the interleaved partition where $A$ and $B$ hold

---

[4]The one-dimensional case is addressed for simplicity, the analysis for a two-dimensional setting is similar. This can also correspond to a one-dimensional signal, e.g. sound or text.

[5]In appendix E we prove upper and lower bounds for a general setting of channel numbers. Furthermore, in appendix G we present simulations which indicate that deviations from the equality stated in theorem 1 are quite rare and unsubstantial in value.

the odd and even input patches, respectively, the multiplicative minimal cut weight will be equal to $\min(r_0^{N/4}, M^{N/2})$ – dependent only on the first layers' channel numbers, and exponential in $N$.[6]

The partitions mentioned above represent two extreme cases that correspond to shortest and longest ranged correlations. However, the min-cut result applies to any partition of the inputs, so that conclusions regarding the layer widths can be established for any intermediate length-scale of correlations. For example, the relevant factors that contribute to the min-cut between $(A, B)$ for which both $A$ and $B$ have contiguous segments of a certain length $\xi$ are $M, r_0, ..., r_{\lceil \log_2 \xi \rceil}$. This is in fact a generalization of the treatment above with $\xi = 1$ for the interleaved partition and $\xi = N/2$ for the left-right partition, and can be understood by flow considerations in the graph underlying the TN: a cut that is located above a certain sub-branch can not assist in cutting the flow between $A$ and $B$ vertices that reside within that sub-branch. Thus, the addition of more parameters to layers $l$ such that $l > \log_2 \xi$ would result in an increase of the capacity of edges in the TN which will not belong to the min-cut.

The observation presented in the previous paragraph has practical implications. For a data-set with features of a characteristic size $D$ (e.g. in a two-dimensional digit classification task, $D$ could be the size of digits that are to be classified), such partitions of length scales $\xi < D$ are guaranteed to separate between different parts of a feature placed in any input location. In order to classify a feature correctly, an elaborate function modeling a strong dependence between different parts of it must be realized by the network. As discussed above, this means that a high measure of entanglement w.r.t. partitions that separate the feature must be supported by the network, and theorem 1 allows us to describe this measure of entanglement in terms of a min-cut in the TN graph. The following 'rule of thumb' is thus implied – the channel numbers up to layer $l = \lceil \log_2 D \rceil$ are more important than those of deeper layers, therefore it is advisable to concentrate more parameters (in the form of more channels) in these levels. Additionally, an analysis of the min-cut in the ConvAC TN shows that among the more important layers $l = 1, ..., \log_2 D$, deeper ones need to be wider, as is apparent for example in the above expression of the minimal cut weight for the high-low partition. In a more general task it may be hard to point out a single most important length scale $D$, however the conclusions presented in this section can be viewed as an incentive to develop adequate means of characterizing the most relevant data correlations for different tasks.

## 6 EXPERIMENTS

The min-cut analysis on the TN representing a deep ConvAC translates prior knowledge on how correlations among input variables (*e.g.* image patches) are modeled, into the architectural design of number of channels per layer in a ConvAC. In this section, we demonstrate empirically that the theoretical findings established above for the deep ConvAC, apply to a regular convolutional network architecture which involves the more common ReLU activations and average or max pooling. Two tasks were designed, one with a short characteristic length to be referred to as the 'local task', and the other with a long characteristic length to be referred to as the 'global task'. Both tasks are based on the MNIST data-set and consist of $64 \times 64$ black background images on top of which resized binary MNIST images were placed in random positions, to make sure we account for correlation distances without biasing towards a particular location in the image. For the local task, the digits were shrunken to $8 \times 8$ images while for the global task they were enlarged to size $32 \times 32$. Note that both tasks are more challenging than the standard MNIST task, and that the local task is even more challenging than the global one.

We designed two network architectures that tackle these two tasks, with a difference in the channel ordering scheme. Each architecture was designed to better match the correlation structure of one of the above tasks, in accordance with the analysis presented in the previous section. In both networks, the first layer is a representation layer – a $3 \times 3$ (stride 1) shared convolutional layer. Following it are 6 hidden layers, each with $1 \times 1$ shared convolution kernels followed by ReLU activations and $2 \times 2$ max pooling (stride 2). Classification in both networks was performed through $Y = 10$ outputs, with prediction following the strongest activation. The difference between the two networks is in the channel ordering – in the *wide-base* (WB) network they are wider in the beginning and narrow down in the deeper layers, while in the *wide-tip* (WT) network they follow the opposite trend. Specifically, we set a parameter $r$ to determine each pair of such networks according to WB: [10; 4r; 4r; 2r; 2r; r; r; 10] and WT: [10; r; r; 2r; 2r; 4r; 4r; 10] (The channel numbers from left to right go from shallow

---

[6]The depth efficiency result shown in Cohen et al. (2016b), can be reproduced by similar graph-theoretic considerations that are related to the exponential dependence in this expression, see appendix F.

to deep). According to the above conclusions, this choice for increase of widths towards deeper layers in the WT network makes it fit the global task in which all layers are important. similarly, the conclusions dictate that the choice WB network makes it fit the local task, in which shallower layers are more important. The specific channel arrangement ensures that the amount of learned parameters for both configurations is equal.

Fig. 4 shows the results of applying both the WB and WT networks to the local and global tasks. Each task consisted of 60000 training images and 10000 test images, in correspondence with the MNIST database. Indeed, the WB network significantly outperforms the WT network in the local task, whereas a clear opposite trend can be seen for the global task. This complies with our theoretical analysis, according to which the WB network which holds more parameters in shallow layers should be able to support short correlation lengths of the input, whereas the WT network in which deeper layers are wider is predicted to put focus on longer correlation lengths. The fact that the global task gets higher accuracies for all choices of $r$ is unsurprising, as it is clearly an easier task.

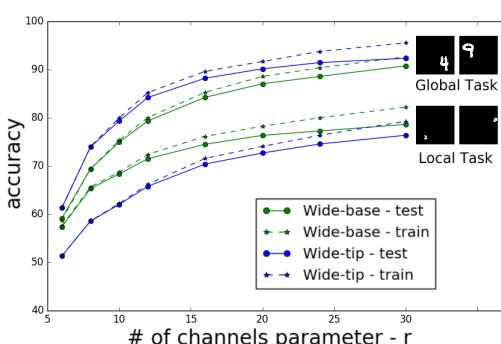

Figure 4: Applying ReLU networks with max pooling to the global and local classification tasks.

Overall, these experiments constitute a demonstration of how prior knowledge regarding a task at hand may be used to tailor the inductive bias of a deep convolutional network by appropriately designing layer widths. We have shown how phenomena that were indicated by the theoretical analysis that was presented in this paper in the context of ConvACs, manifest themselves on the most prevalent and successful convolutional network architectures (ReLU activation, max pooling).[7]

## 7 DISCUSSION

The construction of a deep ConvAC in terms of a TN brought forth the main theoretical achievements of this paper. This method enabled us to carry a graph-theoretic analysis of a convolutional network, and tie its expressiveness to a minimal cut in the graph characterizing it. Our construction began with a structural equivalence between the function realized by a ConvAC and a quantum many-body wave function. This facilitated the transfer of mathematical and conceptual tools employed by physicists, such as the tool of TNs and the concept of 'entanglement measures', providing well-defined quantifiers for a deep network's expressive ability to model correlations between regions of its input. By employing these tools, we were able to present theoretical observations regarding the role that the number of channels in each layer fulfills in the overall expressiveness of a deep convolutional network, and how they affect its ability to model given input correlations. Furthermore, practical implications were presented for the construction of a deep network architecture when there is prior knowledge regarding the input correlations.

Apart from the direct results discussed above, two important interdisciplinary bridges emerge from this work. The results we drew between min-cut in the graph representation of a ConvAC to network expressivity measures, may constitute an initial example for employing the connection to TNs for the application of graph-theoretic measures and tools to the analysis of the function realized by a deep convolutional network. The second bridge, is the mathematical connection between the two fields of quantum physics and deep learning. The field of quantum TNs is a rapidly evolving one, and the established construction of a successful deep learning architecture in the language of TNs may allow applications and insights to be transferred between the two domains. For example, the tree shaped TN that was shown in this work to be equivalent to a deep convolutional network, has been known in the physics community for nearly a decade to be inferior to another deep TN architecture by the name of MERA (Vidal, 2008), in its expressiveness and in its ability to model correlations.

The MERA TN constitutes an exemplar case of how the TNs/deep-learning connection established in this work allows a bi-directional flow of tools and intuition. MERA architecture introduces over-

---

[7]In addition to ReLU ConvNets, we also evaluated ConvACs (results omitted for space). The accuracies were on average 1 percent lower, however the observed trends were the same – an order of 5 percent advantage in favor of WB (WT) network on the local (global) task (respectively).

laps by adding 'disentangling' operations prior to the pooling operations, which, in translation to terms of deep learning, effectively mix activations that are intended to be pooled in different pooling windows. Physicists have a good grasp of how these specific overlapping operations allow a most efficient representation of functions that exhibit high correlations at all length scales (Vidal, 2007). Accordingly, a new view of the role of overlaps in the high expressivity of deep networks as effectively 'disentangling' intricate correlations in the data can be established. In the other direction, as deep convolutional networks are the most empirically successful machine learning architectures to date, physicists may benefit from trading their current 'overlaps by disentangling' scheme to the use of overlapping convolutional windows (proven to contribute exponentially to the expressive capacity of neural networks by Sharir and Shashua (2017)), in their search for expressive representations of quantum wave functions. Overall, We view this work as an exciting bridge for transfer of tools and ideas between fields, and hope it will reinforce a fruitful interdisciplinary discourse.

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

## A    DEFINITION OF MATRICIZATION

An important concept we make use of is *matricization*, which is essentially the rearrangement of a tensor as a matrix. Suppose $\mathcal{A}$ is a tensor of order $N$ and dimension $M_i$ in each mode $i \in [N] := \{1, \ldots, N\}$, and let $(A, B)$ be a partition of $[N]$, *i.e.* $A$ and $B$ are disjoint subsets of $[N]$ whose union gives $[N]$. We may write $A = \{a_1, \ldots, a_{|A|}\}$ where $a_1 < \cdots < a_{|A|}$, and similarly $B = \{b_1, \ldots, b_{|B|}\}$ where $b_1 < \cdots < b_{|B|}$. The *matricization of $\mathcal{A}$ w.r.t. the partition* $(A, B)$, denoted $[\![\mathcal{A}]\!]_{A,B}$, is the $\prod_{t=1}^{|A|} M_{a_t}$-by-$\prod_{t=1}^{|B|} M_{b_t}$ matrix holding the entries of $\mathcal{A}$ such that $\mathcal{A}_{d_1 \ldots d_N}$ is placed in row index $1 + \sum_{t=1}^{|A|} (d_{a_t} - 1) \prod_{t'=t+1}^{|A|} M_{a_{t'}}$ and column index $1 + \sum_{t=1}^{|B|} (d_{b_t} - 1) \prod_{t'=t+1}^{|B|} M_{b_{t'}}$.

## B    INTRODUCTION TO QUANTUM WAVE FUNCTIONS

We provide below a short introduction to the notation used by physicists when describing quantum mechanical properties of a many-body system. We follow relevant derivations in Preskill (1998) and Hall (2013), referring the interested reader to these sources for a more comprehensive mathematical introduction to quantum mechanics.

A state of a system, which is a complete description of a physical system, is given in quantum mechanics as a *ray* in a Hilbert space (to be defined below). Relevant Hilbert spaces in quantum mechanics are vector spaces over the complex numbers. We restrict our discussion to vector spaces over $\mathbb{R}$, as the properties related to complex numbers are not required for our analysis and do not affect it. Physicists denote such vectors in the 'ket' notation, in which a vector $\psi$ is denoted by: $|\psi\rangle \in \mathcal{H}$. The Hilbert space $\mathcal{H}$ has an inner product denoted by $\langle \phi | \psi \rangle$, that maps a pair of two vectors in $\mathcal{H}$ to a scalar. This inner product operation is also referred to as 'projecting $|\psi\rangle$ onto $|\phi\rangle$'. A ray is an equivalence class of vectors that differ by multiplication by a nonzero scalar. For any nonzero ray, a representative of the class, $|\psi\rangle$, is conventionally chosen to have a unit norm: $\langle \psi | \psi \rangle = 1$. A 'bra' notation $\langle \phi |$, is used for the 'dual vector' which formally is a linear mapping between vectors to scalars defined as $|\psi\rangle \mapsto \langle \phi | \psi \rangle$. We can intuitively think of a 'ket' as a column vector and 'bra' as a row vector.

Relevant Hilbert spaces can be infinite dimensional or finite dimensional. We limit our discussion to quantum states which reside in finite dimensional Hilbert spaces, as these lie at the heart of our analogy to convolutional networks. Besides being of interest to us, these spaces are extensively investigated in the physics community as well. For example, the spin component of a spinful particle's wave function resides in a finite dimensional Hilbert space. One can represent a general single particle state $|\psi\rangle \in \mathcal{H}_1$, where $\dim(\mathcal{H}_1) = M$, as a linear combination of some orthonormal basis vectors:

$$|\psi\rangle = \sum_{d=1}^{M} v_d |\psi_d\rangle, \tag{6}$$

where $\mathbf{v} \in \mathbb{R}^M$ is the vector of coefficients compatible with the basis $\{|\psi_d\rangle\}_{d=1}^{M}$ of $\mathcal{H}_1$, each entry of which can be calculated by the projection: $v_d = \langle \psi_d | \psi \rangle$.

The extension to the case of $N$ particles, each with a wave function residing in a local finite dimensional Hilbert space $\mathcal{H}_j$ for $j \in [N]$ (e.g. $N$ spinful particles), is readily available through the tool of a tensor product. In order to define a Hilbert space which is the tensor product of the local Hilbert spaces: $\mathcal{H} := \otimes_{j=1}^{N} \mathcal{H}_j$, we will specify its scalar product. Denote the scalar product in each $\mathcal{H}_j$ by $\langle \cdot | \cdot \rangle_j$, then the scalar product in the tensor product finite dimensional Hilbert space $\mathcal{H} = \otimes_{j=1}^{N} \mathcal{H}_j$ between $|\psi\rangle := \otimes_{j=1}^{N} \left| \psi^{(j)} \right\rangle \in \mathcal{H}$ and $|\phi\rangle := \otimes_{j=1}^{N} \left| \phi^{(j)} \right\rangle \in \mathcal{H}$ is defined by: $\langle \phi | \psi \rangle := \prod_{j=1}^{N} \left\langle \phi^{(j)} | \psi^{(j)} \right\rangle_j, \forall \left| \psi^{(j)} \right\rangle, \left| \phi^{(j)} \right\rangle \in \mathcal{H}_j$.

For simplicity, we set the dimensions of the local Hilbert spaces $\mathcal{H}_j$ to be equal for all $j$, i.e. $\forall j : \dim(\mathcal{H}_j) = M$. In the spin example, this means that the particles have the same spin, e.g. for $N$ electrons (spin $1/2$), $M = 2$. Denoting as above the orthonormal basis of the local Hilbert space by $\{|\psi_d\rangle\}_{d=1}^{M}$, the many-body quantum wave function $|\psi\rangle \in \mathcal{H} = \otimes_{j=1}^{N} \mathcal{H}_j$ can be written as:

$$|\psi\rangle = \sum_{d_1 \ldots d_N = 1}^{M} \mathcal{A}_{d_1 \ldots d_N} |\psi_{d_1}\rangle \otimes \cdots \otimes |\psi_{d_N}\rangle, \tag{7}$$

Reproducing eq. 2.

### B.1    DERIVATION OF THE SCHMIDT DECOMPOSITION

Consider a partition of the above described system of $N$ particles labeled by integers $[N] := \{1, \ldots, N\}$, which splits it into two disjoint subsystems $A \cup B = [N]$ such that $A = \{a_1, \ldots, a_{|A|}\}$ with $a_1 < \ldots < a_{|A|}$

and $B = \{b_1, \ldots, b_{|B|}\}$ with $b_1 < \ldots < b_{|B|}$. Let $\mathcal{H}^A$ and $\mathcal{H}^B$ be the Hilbert spaces in which the many-body wave functions of the particles in subsystems $A$ and $B$ reside, respectively, with $\mathcal{H} = \mathcal{H}^A \otimes \mathcal{H}^B$.[8] The many-body wave function in eq. 2 can be now written as:

$$|\psi\rangle = \sum_{\alpha=1}^{\dim(\mathcal{H}^A)} \sum_{\beta=1}^{\dim(\mathcal{H}^B)} (\llbracket \mathcal{A} \rrbracket_{A,B})_{\alpha,\beta} \left|\psi_\alpha^A\right\rangle \otimes \left|\psi_\beta^B\right\rangle, \tag{8}$$

where $\{\left|\psi_\alpha^A\right\rangle\}_{\alpha=1}^{\dim(\mathcal{H}^A)}$ and $\{\left|\psi_\beta^B\right\rangle\}_{\beta=1}^{\dim(\mathcal{H}^B)}$ are bases for $\mathcal{H}^A$ and $\mathcal{H}^B$, respectively,[9] and $\llbracket \mathcal{A} \rrbracket_{A,B}$ is the matricization of $\mathcal{A}$ w.r.t. the partition $(A, B)$. Let us denote the maximal rank of $\llbracket \mathcal{A} \rrbracket_{A,B}$ by $r := \min(\dim(\mathcal{H}^A), \dim(\mathcal{H}^B))$. A singular value decomposition on $\llbracket \mathcal{A} \rrbracket_{A,B}$ results in the following form (also referred to as the Schmidt decomposition):

$$|\psi\rangle = \sum_{\alpha=1}^{r} \lambda_\alpha \left|\phi_\alpha^A\right\rangle \otimes \left|\phi_\alpha^B\right\rangle, \tag{9}$$

where $\lambda_1 \geq \cdots \geq \lambda_r$ are the singular values of $\llbracket \mathcal{A} \rrbracket_{A,B}$, and $\{\left|\phi_\alpha^A\right\rangle\}_{\alpha=1}^r$, $\{\left|\phi_\alpha^B\right\rangle\}_{\alpha=1}^r$ are $r$ vectors in new bases for $\mathcal{H}^A$ and $\mathcal{H}^B$, respectively, obtained by the decomposition. It is noteworthy that since $|\psi\rangle$ is conventionally chosen to be normalized, the singular values uphold $\sum_\alpha |\lambda_\alpha|^2 = 1$, however this constraint can be relaxed for our needs.

## C    Introduction to Tensor Networks

A *Tensor Network* (TN) is formally represented by an underlying undirected graph that has some special attributes, we elaborate on this formal definition in appendix E.1. In the following, we give a more intuitive description of a TN, which is nonetheless exact and required for our construction of the ConvAC TN. The basic building blocks of a TN are tensors, which are represented by nodes in the network. The order of a tensor represented by a node, is equal to its degree — the number of edges incident to it, also referred to as its legs. Fig. 5(a) shows three examples: 1) A vector, which is a tensor of order 1, is represented by a node with one leg. 2) A matrix, which is a tensor of order 2, is represented by a node with two legs. 3) Accordingly, a tensor of order $N$ is represented in the TN as a node with $N$ legs. In a TN, each edge is associated with a number called the *bond dimension*. The bond dimension assigned to a specific leg of a node, is simply the dimension of the corresponding mode of the tensor represented by this node (see definitions for a mode and its dimension in section 2).

A TN is a collection of such tensors represented by nodes, with edges that can either be connected to a node on one end and loose on the other end or connect between two nodes. Each edge in a TN is represented by an index that runs between 1 and its bond dimension. An index representing an edge which connects between two tensors is called a contracted index, while an index representing an edge with one loose end is called an open index. The set of contracted indices will be denoted by $K = \{k_1, ..., k_P\}$ and the set of open indices will be denoted by $D = \{d_1, ..., d_N\}$. The operation of contracting the network is defined by summation over all of the $P$ contracted indices. The tensor represented by the network, $\mathcal{A}_{d_1 \ldots d_N}$, is of order $N$, *i.e.* its modes correspond to the open indices. Given the entries of the internal tensors of the network, $\mathcal{A}_{d_1 \ldots d_N}$ can be calculated by contracting the entire network.

An example for a contraction of a simple TN is depicted in fig. 5(b). There, a TN corresponding to the operation of multiplying a vector $\mathbf{v} \in \mathbb{R}^{r_1}$ by a matrix $M \in \mathbb{R}^{r_2 \times r_1}$ is performed by summing over the only contracted index, $k$. As there is only one open index, $d$, the result of contracting the network is an order 1 tensor (a vector): $\mathbf{u} \in \mathbb{R}^{r_2}$ which upholds $\mathbf{u} = M\mathbf{v}$. In fig. 5(c) a somewhat more elaborate example is illustrated, where a TN composed of order 2 and 3 tensors represents a tensor of order 5. This network represents a decomposition known as a *tensor train* (Oseledets (2011)) in the tensor analysis community or a *matrix product state* (MPS) (see overview in e.g. Orús (2014)) in the condensed matter physics community, which arranges order 3 tensors in such a 'train' architecture and allows the representation of an order $N$ tensor with a linear (in $N$) amount of parameters. The MPS exemplifies a typical desired quality of TNs. The decomposition of a higher order tensor into a set of sparsely interconnected lower order tensors, was shown (Oseledets and Tyrtyshnikov (2009); Ballani et al. (2013)) to greatly diminish effects related to the curse of dimensionality discussed above.

---

[8]Actually, $\mathcal{H} \cong \mathcal{H}^A \otimes \mathcal{H}^B$ with equality obtained upon a permutation of the local spaces that is compliant with the partition $(A, B)$.

[9]It is possible to write $\left|\psi_\alpha^A\right\rangle = \left|\psi_{d_{a_1}}\right\rangle \otimes \cdots \otimes \left|\psi_{d_{a_{|A|}}}\right\rangle$ and $\left|\psi_\beta^B\right\rangle = \left|\psi_{d_{b_1}}\right\rangle \otimes \cdots \otimes \left|\psi_{d_{b_{|B|}}}\right\rangle$ with some mapping from $\{a_1, \ldots, a_{|A|}\}$ to $\alpha$ and from $\{b_1, \ldots, b_{|B|}\}$ to $\beta$ which corresponds to the matricization formula given in appendix A.

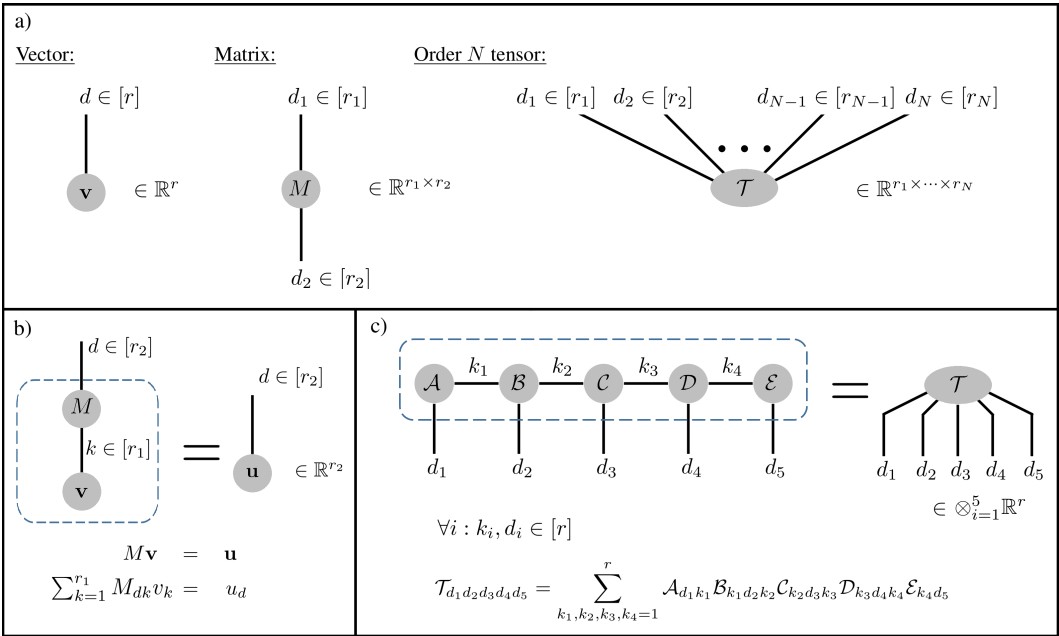

Figure 5: A quick introduction to Tensor Networks. (a) Tensors in the TN are represented by nodes. The degree of the node corresponds to the order of the tensor represented by it. (b) A matrix multiplying a vector in TN notation. The contracted indices are denoted by $k$ and are summed upon. The open indices are denoted by $d$, their number equals the order of the tensor represented by the entire network. All of the indices receive values that range between 1 and their bond dimension. The contraction is marked by the dashed line. (c) A more elaborate example, of a network representing a higher order tensor via contractions over sparsely interconnected lower order tensors. This network is a simple case of a decomposition known as a *tensor train* (Oseledets (2011)) in the tensor analysis community or a *matrix product state* (see overview in e.g. Orús (2014)) in the condensed matter physics community.

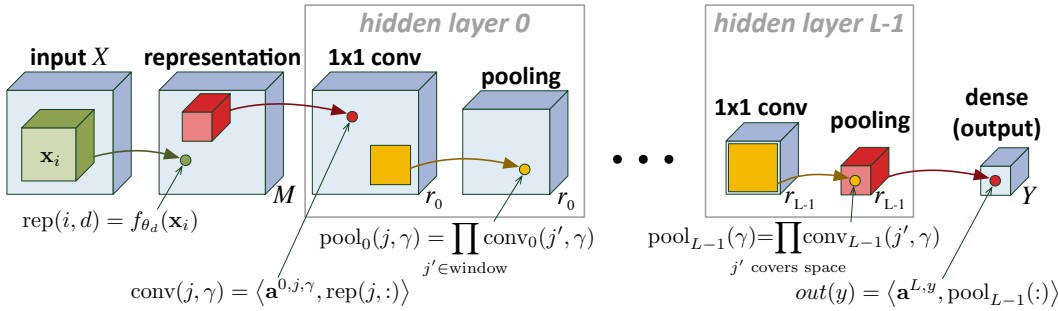

Figure 6: The original Convolutional Arithmetic Circuits network as presented by Cohen et al. (2016b).

# D TENSOR NETWORK CONSTRUCTION OF A CONVAC

We begin by reviewing tensor decompositions of the conv-weights tensor shown in Cohen et al. (2016b) to be equivalent to the shallow and deep versions of the ConvAC network given in the main text and reproduced for convenience in fig. 6.

The CP decomposition of the conv-weights tensor corresponds to a ConvAC depicted in fig. 6 with one hidden layer, which collapses the entire spatial structure through global pooling – a shallow ConvAC. Explicitly, the CP decomposition of the order $N$ conv-weights tensor of a specific class $y$ is a sum of rank-1 tensors, each of which is attained by a tensor product of $N$ weights vectors:

$$\mathcal{A}^y = \sum_{k=1}^{K} v_k^y \cdot \mathbf{a}^{k,1} \otimes \cdots \otimes \mathbf{a}^{k,N}, \tag{10}$$

where $\mathbf{v}^y \in \mathbb{R}^K, \forall y \in [Y]$ and $\mathbf{a}^{k,j} \in \mathbb{R}^M, \forall k \in [K], j \in [N]$.

The deep version of fig. 6, where the pooling windows between convolutional layers are of minimal size, corresponds to a specific tensor decomposition of $\mathcal{A}^y$, which is a restricted version of a *hierarchical Tucker decomposition*, referred to in short as the HT decomposition. The restriction is related to the fact that the pooling scheme of the ConvAC architecture presented in fig. 6 involves only entries from the same channel, while in the general HT decomposition pooling operations would involve entries from different channels. For brevity of notation, we will present the expressions for a scenario where the input patches are aligned along a one-dimensional line (can also correspond to a one-dimensional signal, e.g. sound or text), and the pooling widows are of size 2. The extension to the two-dimensional case follows quite trivially, and was presented in Cohen and Shashua (2017). Under the above conditions, the decomposition corresponding to a deep ConvAC can be defined recursively by (Cohen et al. (2016b)):

$$\phi^{1,j,\gamma} = \sum_{\alpha=1}^{r_0} a_\alpha^{1,j,\gamma} \mathbf{a}^{0,2j-1,\alpha} \otimes \mathbf{a}^{0,2j,\alpha}$$

$$\cdots$$

$$\phi^{l,j,\gamma} = \sum_{\alpha=1}^{r_{l-1}} a_\alpha^{l,j,\gamma} \underbrace{\phi^{l-1,2j-1,\alpha}}_{\text{order } 2^{l-1}} \otimes \underbrace{\phi^{l-1,2j,\alpha}}_{\text{order } 2^{l-1}}$$

$$\cdots$$

$$\mathcal{A}^y = \sum_{\alpha=1}^{r_{L-1}} a_\alpha^{L,y} \underbrace{\phi^{L-1,1,\alpha}}_{\text{order } \frac{N}{2}} \otimes \underbrace{\phi^{L-1,2,\alpha}}_{\text{order } \frac{N}{2}}. \tag{11}$$

The decomposition in eq. 11 recursively constructs the conv-weights tensor $\{\mathcal{A}^y\}_{y \in [Y]}$ by assembling vectors $\{\mathbf{a}^{0,j,\gamma}\}_{j \in [N], \gamma \in [r_0]}$ into tensors $\{\phi^{l,j,\gamma}\}_{l \in [L-1], j \in [N/2^l], \gamma \in [r_l]}$ in an incremental fashion. This is done in the form of tensor products, which are the natural form for tensor decompositions. The index $l$ stands for the level in the decomposition, corresponding to the $l^{th}$ layer of the ConvAC network given in fig. 6. $j$ represents the 'location' in the feature map of level $l$, and $\gamma$ corresponds to the individual tensor in level $l$ and location $j$. The index $r_l$ is referred to as *level-l rank*, and is defined to be the number of tensors in each location of level $l$ (we denote for completeness $r_L := Y$). In the ConvAC network given in fig. 6, $r_l$ is equal to the number of

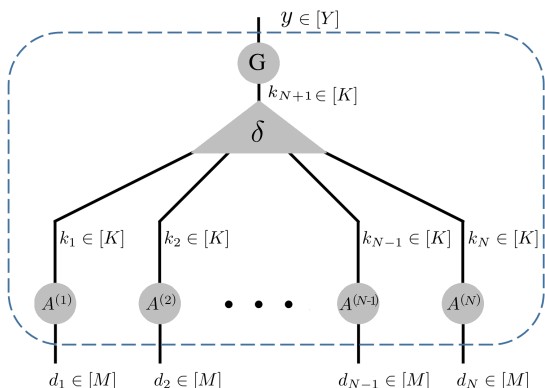

Figure 7: The TN equivalent of the CP decomposition. This is a TN representation of the order $N$ weights tensor $\mathcal{A}_{d_1 \dots d_N}$ underlying the calculation of the ConvAC in fig. 6 in its shallow form, *i.e.* with one hidden layer followed by a global pooling operation which collapses the feature maps into $Y$ different class scores. The matrices $A^{(j)}$ hold the convolutional weights of the hidden layer and the matrix $G$ holds the weights of the final dense layer. The central $\delta$ tensor effectively enforces the same channel pooling, as can be seen by its form in eq. 12 and its role in the calculation of this TN given in eq. 13.

channels in the $l^{th}$ layer — this is important in our analysis of the role played by the channel numbers. The tensor $\phi^{l,j,\gamma}$ is of order $2^l$, and we assume for simplicity that $N$ – the order of $\mathcal{A}^y$, is a power of 2 (this is merely a technical assumption also made in Hackbusch (2012), it does not limit the generality of the analysis). The parameters of the decomposition are the final level weights $\{\mathbf{a}^{L,y} \in \mathbb{R}^{r_{L-1}}\}_{y \in [Y]}$, the intermediate levels' weights $\{\mathbf{a}^{l,j,\gamma} \in \mathbb{R}^{r_{l-1}}\}_{l \in [L-1], j \in [N/2^l], \gamma \in [r_l]}$, and the first level weights $\{\mathbf{a}^{0,j,\gamma} \in \mathbb{R}^M\}_{j \in [N], \gamma \in [r_0]}$.

## D.1 TENSOR NETWORK CONSTRUCTION OF A SHALLOW CONVAC

In order to construct the TN equivalent of the shallow ConvAC, we define the order $N+1$ tensor $\delta \in \mathbb{R}^{K \times \dots \times K}$, referred to as the $\delta$ *tensor*, as follows:

$$\delta_{k_1 \dots k_{N+1}} := \begin{cases} 1, & k_1 = \dots = k_{N+1} \\ 0, & otherwise \end{cases}, \tag{12}$$

with $k_j \in [K] \; \forall j \in [N+1]$, *i.e.* its entries are equal to 1 only on the super-diagonal and are zero otherwise. We shall draw nodes which correspond to such $\delta$ tensors as triangles in the TN, to remind the reader of the restriction given in eq. 12. Let $G \in \mathbb{R}^{Y \times K}$ be a matrix holding the convolutional weight vector of the final layer $\mathbf{v}^y \in \mathbb{R}^K$ in its $y^{th}$ row and let $A^{(j)} \in \mathbb{R}^{K \times M}$ be a matrix holding the convolutional weights vector $\mathbf{a}^{k,j} \in \mathbb{R}^M$ in its $k^{th}$ row. One can observe that per class $y$, the $k^{th}$ summand in eq. 10 is equal to the tensor product of the $N$ vectors residing in the $k^{th}$ rows of all the matrices $A^{(j)}$, $j \in [N]$, multiplied by a final weight associated with class $y$. Tensors represented by nodes in the TN will have parenthesis in the superscript, which denote labels such as the position $j$ in the above, to differentiate them from 'real' indices that must be taken into consideration when contracting the TN. Per convention, such 'real' indices will be written in the subscript.

Having defined the above, the TN equivalent of the CP decomposition is illustrated in fig. 7. Indeed, though they represent the same exact quantity, the form of eq. 10 isn't apparent at a first glance of the network portrayed in fig. 7. Essentially, the TN equivalent of the CP decomposition involves contractions between the matrices $A^{(j)}$, $G$, and the $\delta$ tensor, as can be seen in the expression representing it:

$$\mathcal{A}_{d_1 \dots d_N} = \sum_{k_1, \dots, k_{N+1}=1}^{K} \delta_{k_1 \dots k_{N+1}} A^{(1)}_{k_1 d_1} \cdots A^{(N)}_{k_N d_N} G_{y k_{N+1}}. \tag{13}$$

The role of the $\delta$ tensor in eq. 13 can be observed as 'forcing' elements of the $k^{th}$ row of any matrix $A^{(j)}$ to be multiplied only by elements of $k^{th}$ rows of the other matrices which in effect enforces same channel pooling.[10]

---

[10]If one were to switch the $\delta_{k_1 \dots k_N}$ in eq. 13 by a general tensor $\mathcal{G}_{k_1 \dots k_N} \in \mathbb{R}^{K \times \dots \times K}$, a TN equivalent of an additional acclaimed decomposition would be attained, namely the *Tucker decomposition*. Similar to other tensor decompositions, the Tucker decomposition is more commonly given in an outer product form: $\mathcal{A} = \sum_{k_1, \dots, k_N=1}^{K} \mathcal{G}_{k_1 \dots k_N} \mathbf{a}^{k_1, 1} \otimes \cdots \otimes \mathbf{a}^{k_N, N}$.

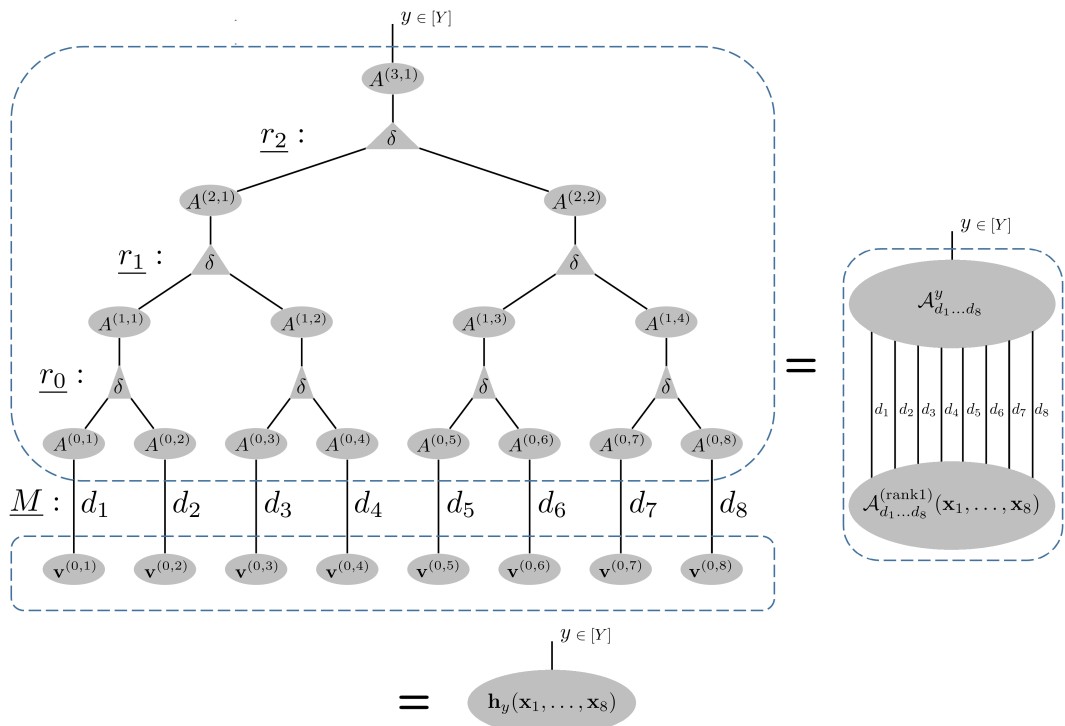

Figure 8: The TN equivalent to the HT decomposition with a same channel pooling scheme corresponding to the calculation of a deep ConvAC given in fig. 6 with $N = 8$. Further details in appendix D.2.

## D.2 TENSOR NETWORK CONSTRUCTION OF A DEEP CONVOLUTIONAL NETWORK

We describe below a TN corresponding to the deep ConvAC calculation, given by eq. 1. The ConvAC calculation is constructed as an inner-product between two tensors: the conv-weights tensor $\mathcal{A}^{y}_{d_1 \ldots d_N}$ which is given in eq. 11 in terms of a tensor decomposition, and $\mathcal{A}^{(\text{rank 1})}_{d_1 \ldots d_N}(\mathbf{x}_1, \ldots, \mathbf{x}_N)$ which is a rank-1 tensor holding the $N \cdot M$ values of the representation layer ($M$ representation functions applied on $N$ input patches).

Fig. 8 displays in full the TN for an $N = 8$ ConvAC calculation. The upper block separated by a dashed line is the TN representing the conv-weights tensor whereas the lower block represents the rank-1 inputs tensor. Considering the upper block, it is worth noting that it is not a sketch of a TN but the actual full description compliant with the graph notations described in appendix C. Accordingly, the two legged nodes represent matrices, where each matrix $A^{(l,j)} \in \mathbb{R}^{r_l \times r_{l-1}}$ (with $r_{-1} := M$) is constructed such that it holds the convolutional weight vector $\mathbf{a}^{l,j,\gamma} \in \mathbb{R}^{r_{l-1}}, \gamma \in [r_l]$ in its $\gamma^{th}$ row. The triangle node appearing between levels $l - 1$ and $l$ represents an order 3 tensor $\delta \in \mathbb{R}^{r_{l-1} \times r_{l-1} \times r_{l-1}}$, obeying eq. 12. The $\delta$ tensor is the element which dictates the same channel pooling in this TN construction.

As mentioned above, the lower block in fig. 8 is the TN representing $\mathcal{A}^{(\text{rank 1})}_{d_1 \ldots d_8}(\mathbf{x}_1, \ldots, \mathbf{x}_8)$. This simple TN is merely a single outer product of $N = 8$ vectors $\mathbf{v}^{(0,j)} \in \mathbb{R}^M, j \in [N]$ composing the representation layer presented in section 3, holding the values $v^{(0,j)}_{d_j} = f_{\theta_{d_j}}(\mathbf{x}_j)$. In compliance with the analogy between the function realized by the ConvAC and the projection of a many-body wave function onto a product state shown in eq. 4, the form $\mathcal{A}^{(\text{rank 1})}_{d_1 \ldots d_8}$ assumes is exactly the form that the coefficients tensor of a product state assumes when represented as a TN. As can be seen in fig. 8, a final contraction of the indices $d_1, \ldots, d_8$ results in the class scores vector calculated by the ConvAC, $\mathbf{h}_y(\mathbf{x}_1, \ldots, \mathbf{x}_8)$.

The calculation performed by a one-dimensional ConvAC for a general $N$ (s.t. $\log_2 N \in \mathbb{N}$), is given by the recursively defined TN representation shown in fig. 9. $\mathbf{v}^{(l,j)} \in \mathbb{R}^{r_{l-1}}$ is a vector of actual activations generated during a computation across in the $l^{th}$ level of the network shown in fig. 6. Recall that $r_{-1} := M$, and that $\mathbf{v}^{(0,j)} \in \mathbb{R}^M, j \in [N]$ is a vector in the representation layer (see fig. 8). To demonstrate that this TN indeed defines the calculations performed by a ConvAC, we conjecture that the equality in fig. 9 holds, namely that for

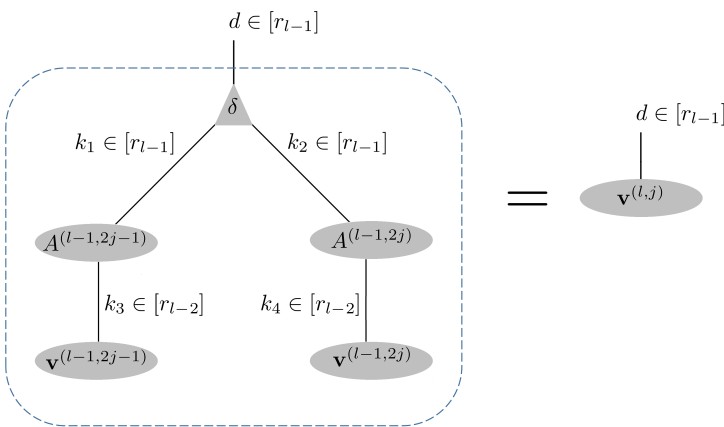

Figure 9: A recursive building block of the deep ConvAC TN. This scheme is the TN equivalent of two feature vectors in the $l-1$ level being operated on with the conv→pool sequence of a deep ConvAC shown in fig. 6, as is demonstrated below.

$l = 1, ..., L = \log_2 N$ the $d^{th}$ component of each such activations vector is given in terms of the vectors in the preceding layer by:

$$v_d^{(l,j)} = \sum_{k_1,k_2=1}^{r_l-1} \sum_{k_3,k_4=1}^{r_l-2} A_{k_1 k_3}^{(l-1,2j-1)} v_{k_3}^{(l-1,2j-1)} A_{k_2 k_4}^{(l-1,2j)} v_{k_4}^{(l-1,2j)} \delta_{k_1 k_2 d}$$

$$= \sum_{k_1,k_2=1}^{r_l-1} (A^{(l-1,2j-1)} \mathbf{v}^{(l-1,2j-1)})_{k_1} (A^{(l-1,2j)} \mathbf{v}^{(l-1,2j)})_{k_2} \delta_{k_1 k_2 d}, \tag{14}$$

where $d \in [r_{l-1}]$. In the first line of eq. 14 we simply followed the TN prescription given in appendix C and wrote a summation over all of the contracted indices in the left hand side of fig. 9, and in the second line we used the definition of matrix multiplication. According to the construction of $A^{(l,j)}$ given in appendix D.2, the vector $\mathbf{u}^{(l,j)} \in \mathbb{R}^{r_l}$ defined by $\mathbf{u}^{(l,j)} := A^{(l,j)} \mathbf{v}^{(l,j)}$, upholds $u_\gamma = \left\langle \mathbf{a}^{l,j,\gamma}, \mathbf{v}^{(l,j)} \right\rangle$, $\gamma \in [r_l]$ where the weights vector $\mathbf{a}^{l,j,\gamma} \in \mathbb{R}^{r_{l-1}}$ was introduced in eq. 11 . Thus, eq. 14 is reduced to:

$$v_d^{(l,j)} = \sum_{k_1,k_2=1}^{r_l-1} \left\langle \mathbf{a}^{l-1,2j-1,k_1}, \mathbf{v}^{(l-1,2j-1)} \right\rangle \left\langle \mathbf{a}^{l-1,2j,k_2}, \mathbf{v}^{(l-1,2j)} \right\rangle \delta_{k_1 k_2 d}. \tag{15}$$

Finally, by definition of the $\delta$ tensor, the sum vanishes and we obtain the required expression for the operation of the ConvAC:

$$v_d^{(l,j)} = \left\langle \mathbf{a}^{l-1,2j-1,d}, \mathbf{v}^{(l-1,2j-1)} \right\rangle \left\langle \mathbf{a}^{l-1,2j,d}, \mathbf{v}^{(l-1,2j)} \right\rangle, \tag{16}$$

where an activation in the $d^{th}$ feature map of the $l^{th}$ level holds the multiplicative pooling of the results of two activation vectors from the previous layer convolved with the $d^{th}$ convolutional weight vector for that layer. Applying this procedure recursively is exactly the conv→pool→ ... →conv→pool scheme that lies at the heart of the ConvAC operation (fig. 6). Recalling that $r_L := Y$, the output of the network is given by:

$$\mathbf{h}_y(\mathbf{x}_1..., \mathbf{x}_N) = A^{(L,1)} \mathbf{v}^{(L,1)}. \tag{17}$$

To conclude this section, we have presented a translation of the computation performed by a ConvAC to a TN. The convolutional weights are arranged as matrices (two legged nodes) placed along the network, and the same channel pooling characteristic is made available due to three legged $\delta$ tensors in a deep network, and an $N+1$ legged $\delta$ tensor in a shallow network. Finally, and most importantly for our analysis, the bond dimension of each level in the TN representing the ConvAC is equal to $r_l$, which is the number of feature maps (*i.e.* the number of channels) comprising that level in the corresponding ConvAC architecture.

## E    PROOF OF MAIN RESULT

Below we provide upper and lower bounds on the ability of a deep ConvAC to model correlations of its inputs, as measured by the Schmidt entanglement measure (see section 4 for definition). We address a general setting

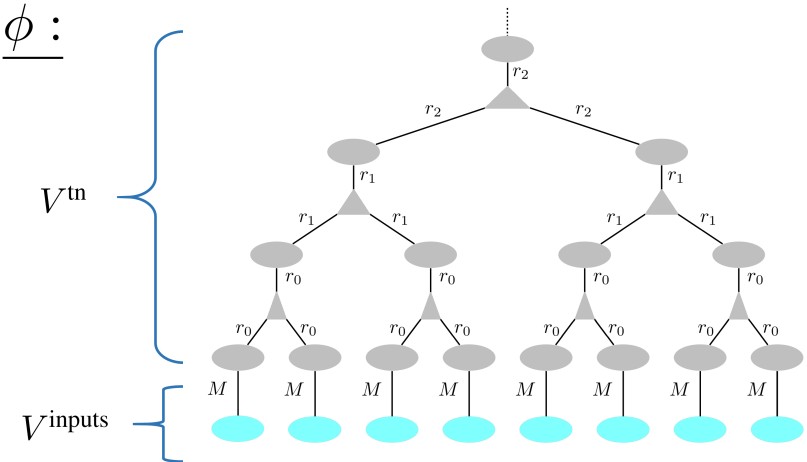

Figure 10: The components comprising a 'ConvAC-weights TN' $\phi$ that describes the weights tensor $\mathcal{A}^y$ of a ConvAC, are an undirected graph $G(V, E)$ and a bond dimensions function $c$. The bond dimension is specified next to each edge $e \in E$, and is given by the function $c(e)$. As shown in appendix D.2, the bond dimension of the edges in each layer of this TN is equal to the number of channels in the corresponding layer in the ConvAC. The node set in the graph $G(V, E)$ presented above decomposes to $V = V^{\text{tn}} \cup V^{\text{inputs}}$, where $V^{\text{tn}}$ (grey) are vertices which correspond to tensors in the ConvAC TN and $V^{\text{inputs}}$ (blue) are degree 1 vertices which correspond to the $N$ open edges in the ConvAC TN. The vertices in $V^{\text{inputs}}$ are 'virtual' — were added for completeness, so $G$ can be viewed as a legal graph. The open edge emanating from the top-most tensor (marked by a dashed line) is omitted from the graph, as it does not effect our analysis below — no flow between any two input groups can pass through it.

of the number of channels in a deep ConvAC. The result stated in theorem 1, which applies when all of the channel numbers in a deep ConvAC architecture are powers of some integer, is implied (specifically by the equality of the upper bound in claim 1 and the lower bound in lemma 2 below). We begin by presenting a description of the TN as a 'legal' graph in section E.1 and move on to prove the bounds in sec E.2.

## E.1    THE CONVAC TENSOR NETWORK AS A GRAPH

The ability to represent a deep convolutional network (ConvAC) as a 'legal' graph, is a key accomplishment that the Tensor Networks tool brings forth. Our main results rely on this graph-theoretic description and tie the expressiveness of a ConvAC to a minimal cut in the graph characterizing it, via the connection to quantum entanglement measures. This is in fact a utilization of the 'Quantum min-cut max-flow' concept presented by Cui et al. (2016). Essentially, the quantum max-flow between $A$ and $B$ is a measure of the ability of the TN to model correlations between $A$ and $B$, and the quantum min-cut is a quantity that bounds this ability and can be directly inferred from the graph defining it — that of the corresponding TN.

We focus on the TN that describes $\mathcal{A}^y_{d_1 \ldots d_N}$, which is the upper block of fig. 8 and is also reproduced as a stand-alone TN in fig. 10, referred to as the 'ConvAC-weights TN' and denoted by $\phi$. The TN $\phi$ has $N$ open edges with bond dimension $M$ that are to be contracted with the inputs $\mathbf{v}^{(0,j)} \in \mathbb{R}^M$, $j \in [N]$ and one open edge with bond dimension $Y$ representing the values $\mathcal{A}^y_{d_1 \ldots d_N}$, $y \in [Y]$ upon such a contraction, as is shown in fig 8.

To turn $\phi$ into a graph we do the following. First, we remove the open edge associated with the output. As our analysis is going to be based on flow between groups of input vertices, no flow can pass through that open edge therefore removing it does not influence our analysis. Second, we add $N$ virtual vertices incident to the open edges associated with the input. Those virtual vertices are the only vertices whose degree is equal to 1 (see fig. 10). The TN $\phi$ is now described below using graph terminology:

- An undirected graph $G(V, E)$, with a set of vertices $V$ and a set of edges $E$. The set of nodes is divided into two subsets $V = V^{\text{tn}} \cup V^{\text{inputs}}$, where $V^{\text{inputs}}$ are the $N$ degree-1 virtual vertices and $V^{\text{tn}}$ corresponds to tensors of the TN.

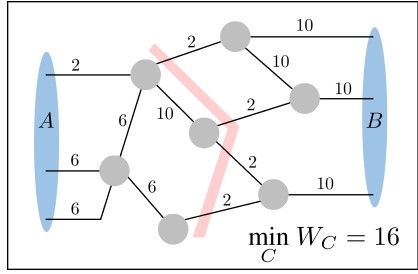

Figure 11: An example for the minimal multiplicative cut between $A$ and $B$ in a simple TN.

- A function $c : E \rightarrow \mathbb{N}$, associating a number $r \in \mathbb{N}$ with each edge in the graph, that equals to the bond dimension of the corresponding edge in the TN.

Having described the object representing the ConvAC-weights TN $\phi$, let us define an edge-cut set with respect to a partition of the $N$ nodes of $V^{\text{inputs}}$, and then introduce a cut weight associated with such a set. An edge-cut set with respect to the partition $V^A \cup V^B = V^{\text{inputs}}$ is a set of edges $C$ s.t. there exists a partition $\tilde{V}^A \cup \tilde{V}^B = V$ with $V^A \subset \tilde{V}^A$ , $V^B \subset \tilde{V}^B$, and $C = \{(u,v) \in E : u \in \tilde{V}^A, v \in \tilde{V}^B\}$. We note that this is a regular definition of an edge-cut set in a graph $G$ with respect to the partition of vertices $(V^A, V^B)$. Let $C = \{e_1, ..., e_{|C|}\}$ be such a set, we define its multiplicative cut weight as:

$$W_C = \prod_{i=1}^{|C|} c(e_i). \tag{18}$$

The weight definition given in eq. 18 is simply a multiplication of the bond dimensions of all the edges in a cut. Fig. 11 shows a pictorial demonstration of this weight definition, which is at the center of our results to come. In the following section, we use a max-flow / min-cut analysis on $\phi$ to obtain new results on the expressivity of the corresponding deep ConvAC via measures of entanglement w.r.t. a bi-partition of its input patches, which are related to the number of channels in each layer of the ConvAC.

## E.2 Bounds on the Entanglement Measure

In claim 1 below, we provide an upper bound on the ability of a deep ConvAC to model correlations of its inputs, as measured by the Schmidt entanglement measure (see section 4). This claim is closely related to attributes of TNs that are known in different forms in the literature.

**Claim 1.** *Let $(A, B)$ be a partition of $[N]$, and $[\![\mathcal{A}^y]\!]_{A,B}$ be the matricization w.r.t. $(A, B)$ of a conv-weights tensor $\mathcal{A}^y$ realized by a ConvAC depicted in fig. 6 with pooling window of size 2 (the deep ConvAC network). Let $G(V, E, c)$ be the graph representation of $\phi$ corresponding to the ConvAC-weights TN, and let $(V^A, V^B)$ be the degree 1 vertices partition in $G$ corresponding to $(A, B)$. Then, the rank of the matricization $[\![\mathcal{A}^y]\!]_{A,B}$ is no greater than: $\min_C W_C$, where $C$ is a cut w.r.t $(V^A, V^B)$ and $W_C$ is the multiplicative weight defined by eq. 18.*

*Proof.* We will use the example shown in fig. 12(a) of a general TN with arbitrary connectivity. The edges of the TN $\phi$ are marked by the index associated with them. Any index $p \in \{d, k\}$ runs between 1 and its bond dimension marked by $c_p$, which upholds $c_p := c(e_p)$ where $e_p \in E$ is the edge associated with the index $p$. For the given partition $(A, B)$, denote $A = \{a_1, ..., a_{|A|}\}$ , $B = \{b_1, ..., b_{|B|}\}$ and let $I_A \cup I_B = \{d_1, \ldots, d_N\}$ be the corresponding partition of external indices, where $I_A = \{d_{a_1}, ..., d_{a_{|A|}}\}$ and $I_B = \{d_{b_1}, ..., d_{b_{|B|}}\}$. Let $\mathcal{H}^A$ and $\mathcal{H}^B$ be the spaces corresponding to the different configurations of the indices in $I_A$ and $I_B$, respectively, their dimensions given by:

$$\dim(\mathcal{H}^A) = \prod_{i=1}^{|A|} c_{d_{a_i}} , \ \dim(\mathcal{H}^B) = \prod_{i=1}^{|B|} c_{d_{b_i}}. \tag{19}$$

In the example shown in fig. 12(a), the graph is arranged s.t. $A$ is on the left and $B$ is on the right. The marked cut $C$ that separates between $A$ and $B$ is arbitrarily chosen as a representative cut, and we denote the indices of the cut edges by $I_C = \{k_1, ..., k_{|C|}\}$. It is noteworthy, that any index $k_i$ in the cut is allowed to be an external index, *i.e.* the cut is allowed to contain any amount of external edges.

Now, two contractions can be performed, separately contracting all the tensors to the left of the cut and to the right of it. We are left with two higher order tensors, $\mathcal{X}_{d_{a_1}...d_{a_{|A|}} k_1...k_{|C|}}$ and $\mathcal{Y}_{k_1...k_{|C|} d_{b_1}...d_{b_{|B|}}}$ each with

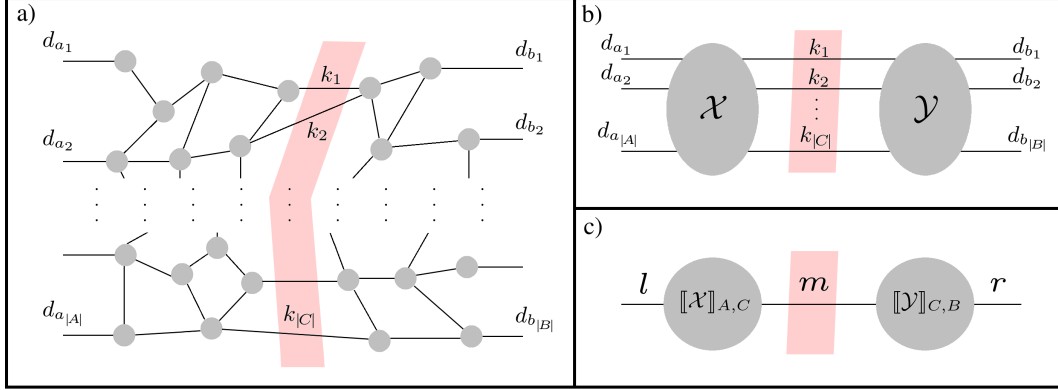

Figure 12: Accompanying illustrations for the proof of claim 1. (a) An example for an arbitrarily inter-connected TN with $N$ external indices, arranged such that the indices corresponding to group $A$ are on the left and indices corresponding to group $B$ are on the right. The cut marked in pink in the middle separates between $A$ and $B$. (b) A contraction of all the internal indices to the left and to the right of the cut results in two higher order tensors, each with external indices only from group $A$ or $B$, connected to each other by the edges of the cut. (c) Finally coalescing the indices into three groups, results in a matrix that on one hand is equal to the matricization w.r.t. $(A, B)$ of the tensor represented by a TN in (a), and on the other is equal to a multiplication of matrices, the rank of which is upper bounded by $\prod_{i=1}^{|C|} c_{k_i}$, thus proving claim 1.

external indices only from $I_A$ or $I_B$, connected to each other by the edges of the cut, as is depicted in fig. 12(b). If any cut index $c_i$ is equal to any external index $d_j$, then respective tensor simply includes the term $\delta_{c_i d_j}$.

Note that the space corresponding to the different configurations of the cut indices $I_C$ is of dimensions $\prod_{i=1}^{|C|} c_{k_i}$, which is exactly equal to $W_C$ (see eq. 18), since by definition $c_{k_i} = c(e_{k_i})$. Next, coalescing the indices in $I_A$ into a single index representing all of the external indices to the left of the network: $l \in [\dim(\mathcal{H}^A)]$, the indices in $I_B$ into a single index representing all of the external indices to the right of the network: $r \in [\dim(\mathcal{H}^B)]$, and the indices in $I_C$ into a single index representing all of the cut indices: $m \in [W_C]$, a TN which is equal to the matricization $[\![\mathcal{A}]\!]_{A,B}$ is obtained (fig. 12(c)).

According to the TN in fig. 12(c), the matricization $[\![\mathcal{A}]\!]_{A,B}$ can be written as a multiplication of two matrices. Component wise, this can be written as:

$$([\![\mathcal{A}]\!]_{A,B})_{lr} = \sum_{m=1}^{W_C} ([\![\mathcal{X}]\!]_{A,C})_{lm}([\![\mathcal{Y}]\!]_{C,B})_{mr}, \tag{20}$$

where any amount of cut indices that are also external indices translate as blocks of the identity matrix on the diagonal. Finally, since this construction is true for any cut $C$ w.r.t $(A, B)$, the rank of $[\![\mathcal{A}]\!]_{A,B}$ upholds: $\text{rank}([\![\mathcal{A}]\!]_{A,B}) \leq \min_C W_C$, satisfying the claim for any general TN, and specifically for the ConvAC TN. $\square$

The upper bound provided above, alerts us when a deep ConvAC is too weak to model a desired correlation structure, according to the number of channels in each layer. Below, we provide a lower bound similar in spirit to a bound shown in Cui et al. (2016). Their claim is applicable for a TN with general tensors (no $\delta$ tensors), and we adapt it to the ConvAC-weights TN (that has $\delta$ tensors) which in effect ensures us that the entanglement measure cannot fall below a certain value for any specific arrangement of channels per layer.

**Theorem 2.** *Let $(A, B)$ be a partition of $[N]$, and $[\![\mathcal{A}^y]\!]_{A,B}$ be the matricization w.r.t. $(A, B)$ of a conv-weights tensor $\mathcal{A}^y$ realized by a ConvAC depicted in fig. 6 with pooling window of size 2 (the deep ConvAC network). Let $G(V, E, c)$ the graph representation of $\phi$ corresponding to the ConvAC-weights TN, and let $(V^A, V^B)$ be the degree 1 vertices partition in $G$ corresponding to $(A, B)$.*

*Let $\phi^p$ be the TN represented by $G(V, E, c^p)$ where $\forall e : c^p(e) := \max_{n \in \mathbb{N}} p^n$ s.t. $p^n \leq c(e)$. In words, $\phi^p$ is a TN with the same connectivity as $\phi$, where all of the bond dimensions are modified to be equal the closest power of $p$ to their value in $\phi$ from below. Let $W_C^p$ be the weight of a cut $C$ w.r.t. $(V^A, V^B)$ in the network $\phi^p$. Then, the rank of the matricization $[\![\mathcal{A}^y]\!]_{A,B}$ is at least: $\max_p \min_C W_C^p$ almost always, i.e. for all configurations of the ConvAC network weights but a set of Lebesgue measure zero.*

Theorem 2 above implies that the upper bound given in Claim 1 is saturated when all of the channel numbers in a deep ConvAC architecture are powers of some integer $p$. For a general arrangement of channel numbers, the upper bound is not tight and theorem 2 guarantees that the rank will not be lower than that of any ConvAC architecture with channel numbers which are powers of some integer $p$ yet are not higher than the original ConvAC channel numbers. Even though this is the lower bound we prove, we have a reason to believe the actual lower bound is much tighter. In section G, we show simulations which indicate that deviations from the upper bound are actually quite rare and unsubstantial in value.

In the following we prove theorem 2. Our proof strategy is similar to the one taken in Cui et al. (2016), however we must deal with the restricted $\delta$ tensors present in the network corresponding to a ConvAC (the triangle nodes in fig. 8). We first quote and show a few results that will be of use to us. We begin by quoting a claim regarding the prevalence of the maximal matrix rank for matrices whose entries are polynomial functions — claim 2. Next, we quote a famous graph theory result known as the Undirected Menger's Theorem (Menger (1927), Elias et al. (1956), Ford and Fulkerson (1956)) which relates the number of edge disjoint paths in an undirected graph to the cardinality of the minimal cut — theorem 3. After this, we show that the rank of matricization of the tensor represented by $\phi^p$ that is defined in theorem 2, is a lower bound on the rank of matricization of the tensor represented by $\phi$ — lemma 1. Then, we prove that the upper bound in claim 1 is tight when all of the channel numbers are any powers of the same integer $p \in \mathbb{N}$ — lemma 2 (effectively showing theorem 1). Finally, when all the preliminaries are in place, we show how the result in theorem 2 is implied.

**Claim 2.** *Let $M, N, K \in \mathbb{N}$, $1 \leq r \leq \min\{M, N\}$ and a polynomial mapping $A : \mathbb{R}^K \to \mathbb{R}^{M \times N}$, i.e. for every $i \in [M]$ and $j \in [N]$ it holds that $A_{ij} : \mathbb{R}^K \to \mathbb{R}$ is a polynomial function. If there exists a point $\mathbf{x} \in \mathbb{R}^K$ s.t. $rank(A(\mathbf{x})) \geq r$, then the set $\{\mathbf{x} \in \mathbb{R}^K : rank(A(\mathbf{x})) < r\}$ has zero measure (w.r.t. the Lebesgue measure over $\mathbb{R}^K$).*

*Proof.* See Sharir et al.. □

Claim 2 implies that it suffices to show an assignment of the ConvAC network weights achieving a certain rank of matricization of the conv-weights tensor, in order to show this is the rank for all configurations of the network weights but a set of Lebesgue measure zero. Essentially, this means that it is enough to provide a specific assignment that achieves the required bound in theorem 2 in order to prove the theorem. Next, we present the following well-known graph theory result:

**Theorem 3.** *(Menger (1927), Elias et al. (1956), Ford and Fulkerson (1956)) [Undirected Menger's Theorem] Let $G = (V, E)$ be an undirected graph with a specified partition $(A, B)$ of the set of degree 1 vertices. Let $MF(G)$ be the maximum number of edge disjoint paths (paths which are allowed to share vertices but not edges) in $G$ connecting a vertex in $A$ to a vertex in $B$. Let $MC(G)$ be the minimum cardinality of all edge-cut sets between $A$ and $B$. Then, $MF(G) = MC(G)$.*

*Proof.* See e.g. Cui et al. (2016). □

Theorem 3 will assist us in the proof of lemma 2. We will use it in order to assert the existence of edge disjoint paths in an auxiliary graph (fig. 13), which we eventually utilize in order to provide the required weights assignment in lemma 2. Next, we show lemma 1, which roughly states that a tensor which 'contains' another tensor in some sense will not have a lower matricization rank than that of the 'contained' tensor.

**Lemma 1.** *Let $(A, B)$ be a partition of $[N]$, and $[\![\mathcal{A}^y]\!]_{A,B}$ be the matricization w.r.t. $(A, B)$ of a conv-weights tensor $\mathcal{A}^y$ realized by a ConvAC depicted in fig. 6. Let $\phi$ be the TN corresponding to this ConvAC network, and let $\phi^p$ be a TN with the same connectivity as $\phi$, where all of the bond dimensions are modified to be equal the closest power of $p$ to their value in $\phi$ from below. Let $(\mathcal{A}^p)^y$ be the tensor represented by $\phi^p$ and let there exist an assignment of all of the tensors in the network $\phi^p$ for which $rank([\![(\mathcal{A}^p)^y]\!]_{A,B}) = R$. Then, $rank([\![\mathcal{A}^y]\!]_{A,B})$ is at least $R$ almost always, i.e. for all configurations of the weights of $\phi$ but a set of Lebesgue measure zero.*

*Proof.* Consider the specific assignment of all of the tensors in the network $\phi^p$ which achieves $rank([\![(\mathcal{A}^p)^y]\!]_{A,B}) = R$, and leads to the resultant tensor $(\mathcal{A}^p)^y$ upon contraction of the network. Observing the form of the deep ConvAC TN presented in appendix D.2, we see that it is composed of $\delta$ tensors and of weight matrices $A^{(l,j)} \in \mathbb{R}^{r_l \times r_{l-1}}$. Recalling that the entries of the former are dictated by construction and obey eq. 12, the assignment of all of the tensors in the network $\phi^p$ is an assignment of all entries of the weight matrices in $\phi^p$ denoted by $(A^p)^{(l,j)}$, $l \in [L] \vee \{0\}$, $j \in [N/2^l]$.

We denote the bond dimension at level $l \in [L] \vee \{-1, 0\}$ of $\phi^p$ by $r_l^p$ (recall that we defined $r_{-1} = M$). By the definition of $\phi^p$, this bond dimension cannot be higher than the bond dimension in the corresponding level in $\phi$ : $\forall l \ r_l^p \leq r_l$. Accordingly, the matrices in $\phi$ do not have lower dimensions (rows or columns) than the corresponding matrices in $\phi^p$. Thus, one can choose an assignment of the weights of all the matrices in $\phi$

to uphold the given assignment for the matrices in $\phi^p$ in their upper left blocks, and assign zeros in the extra spaces:

$$(A^{(l,j)})_{i_1 i_2} = \begin{cases} ((A^p)^{(l,j)})_{i_1 i_2}, & i_1 \leq r_l^p, i_2 \leq r_{l-1}^p \\ 0, & otherwise \end{cases} . \tag{21}$$

Next, we consider a contraction of all the internal indices of $\phi$, which by definition results in the conv-weights tensor $\mathcal{A}^y$. In this contraction, one can split the sum over all of the indices that range in $[r_l]$ for which $r_l^p < r_l$ into two sums: one ranging in $[r_l^p]$ and the other in $r_l^p + [r_l - r_l^p]$. For clarity we will not provide an expression for the entire contraction of $\phi$ which involves many internal indices. To understand the sum splitting schematically, let $k_l$ be an index that ranges in $[r_l]$ for which $r_l^p < r_l$. We perform the following splitting on the sum over $k_l$:

$$\sum_{k_l=1}^{r_l} \{\cdots\} \quad \to \quad \sum_{k_l^{\text{low}}=1}^{r_l^p} \{\cdots\} \;+\; \sum_{k_l^{\text{high}}=r_l^p+1}^{r_l} \{\cdots\}, \tag{22}$$

where $k_l$ is switched into $k_l^{\text{high/low}}$ in all of the summands in the respective sums. The overall contraction will now be split into many sums involving different 'high' and 'low' indices. According to the assignment of $A^{(l,j)}$ (eq. 21), all sums that include any index labeled by 'high' will vanish, and we will be left with a single contraction sum over all the indices labeled by 'low'. It is important to note that a $\delta$ tensor of dimension $r_l$ provides that same values as a $\delta$ tensor of dimension $r_l^p$ when observing only its first $r_l^p$ entries in each dimension, as is clear from the $\delta$ tensor definition in eq. 12. Finally, we observe that this construction leads to $\mathcal{A}^y$ containing the tensor $(\mathcal{A}^p)^y$ as a hypercube in its entirety and holding zeros elsewhere, leading to $\text{rank}(\llbracket \mathcal{A}^y \rrbracket_{A,B}) = \text{rank}(\llbracket (\mathcal{A}^p)^y \rrbracket_{A,B}) = R$. Using claim 2, this specific assignment implies that $\text{rank}(\llbracket \mathcal{A}^y \rrbracket_{A,B})$ is at least $R$ for all configurations of the weights of $\phi$ but a set of Lebesgue measure zero, satisfying the lemma. $\qquad\square$

Lemma 1 basically implies that showing that the upper bound on the rank of the matricization of the deep ConvAC conv-weights tensor that is presented in claim 1 is tight when all of the channel numbers are powers of some integer $p$ (which we show below in lemma 2), is enough in order to prove the lower bound stated in theorem 2.

**Lemma 2.** *Let $(A, B)$ be a partition of $[N]$, and $\llbracket \mathcal{A}^y \rrbracket_{A,B}$ be the matricization w.r.t. $(A, B)$ of a conv-weights tensor $\mathcal{A}^y$ realized by a ConvAC depicted in fig. 6 with pooling window of size $2$ (the deep ConvAC network). Let $G(V, E, c)$ represent the TN $\phi$ corresponding to the ConvAC network s.t. $\forall e \in E, \exists n \in \mathbb{N} : c(e) = p^n$ , and let $(V^A, V^B)$ be the vertices partition of $V^{inputs}$ in $G$ corresponding to $(A, B)$. Let $W_C$ be the weight of a cut $C$ w.r.t. $(V^A, V^B)$. Then, the rank of the matricization $\llbracket \mathcal{A}^y \rrbracket_{A,B}$ is at least $\min_C W_C$ almost always, i.e. for all configurations of the ConvAC network weights but a set of Lebesgue measure zero.*

It is noteworthy, that lemma 2 is stated similarly to claim 1, with two differences: **1)** $\min_C W_C$ appears as a lower bound on the rank of matricization of the conv-weights tensor rather than an upper bound, and **2)** all of the channel numbers are restricted to powers of the same integer $p$. That is to say, by proving this lemma we in fact show that the upper bound proven in claim 1 is tight for this quite general setting of channel numbers.

*Proof.* (of lemma 2)

In the following, we provide an assignment of indices for the tensors in $\phi$ for which the rank of the matricization $\llbracket \mathcal{A}^y \rrbracket_{A,B}$ is at least: $\min_C W_C$. In accordance with claim 2, this will satisfy the lemma as it implies this rank is achieved for all configurations of the ConvAC network weights but a set of Lebesgue measure zero.

The proof of lemma 2 is organized as follows. We begin with the construction of the TN $\phi^*$ presented in fig. 13 from the original network $\phi$, and the show that it suffices to analyze $\phi^*$ for our purposes. Next, we elaborate on the form that the $\delta$ tensors in $\phi$ assume when constructed in $\phi^*$. We then use this form to define the concept of $\delta$ restricted edge disjoint paths, which morally are paths from $A$ to $B$ that are guaranteed to be compliant with the form of a $\delta$ tensor when passing through it. Finally, we use such paths in order to provide an assignment of the indices for the tensors in $\phi^*$ which upholds the required $\delta$ condition.

### $\phi^*$ and the Equivalence of Ranks:

For the given partition $(A, B)$, denote $A = \{a_1, ..., a_{|A|}\}$, $B = \{b_1, ..., b_{|B|}\}$ and let $I_A \cup I_B = \{d_1, \ldots, d_N\}$ be the corresponding partition of external indices, where $I_A = \{d_{a_1}, ..., d_{a_{|A|}}\}$ and $I_B = \{d_{b_1}, ..., d_{b_{|B|}}\}$. Let $\mathcal{H}^A$ and $\mathcal{H}^B$ with dimensions obeying eq. 19 be the spaces corresponding to the different configurations of the indices in $I_A$ and $I_B$, respectively. We construct a TN $\phi^*$ with a graph $G^*(V^*, E^*)$ and a bond dimensions function $c^* : E^* \to \mathbb{N}$ for which there is a one-to-one correspondence between the tensor assignments in $\phi$

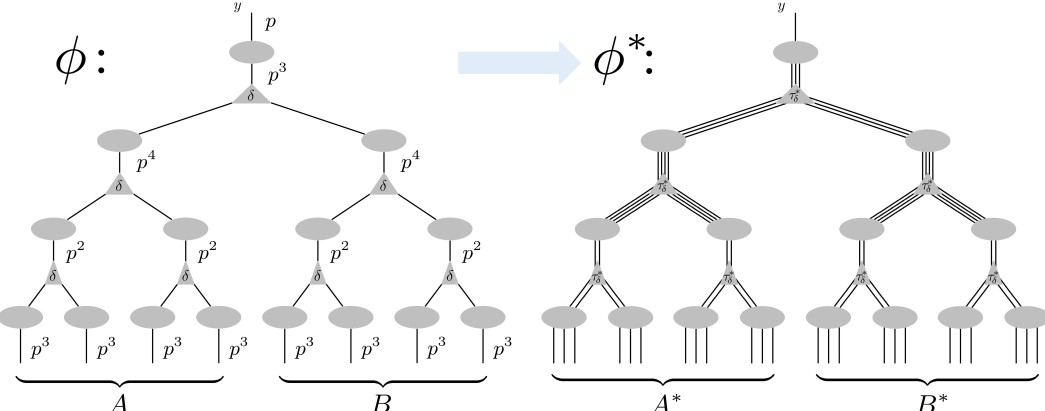

Figure 13: An example for the construction of the TN $\phi^*$ out of the original TN $\phi$ which represents a deep ConvAC (section. D.2), in the case where all of the bond dimensions are powers of some integer number $p$. $n_e$ edges with bond dimension $p$ are placed in $\phi^*$ in the position of each edge $e$ in $\phi$ that has a bond dimension $p^{n_e}$. This construction preserves the value of the minimal multiplicative cut between any two groups of external indices, $(A, B)$ in $\phi$ (here chosen as the left-right partition for example) which correspond to $(A^*, B^*)$ in $\phi^*$.

and tensor assignments in $\phi^*$, such that the resulting linear maps between $\mathcal{H}^A$ and $\mathcal{H}^B$ have the same rank. For each edge $e \in E$, denote $n_e := \log_p c(e)$. By the conditions of the lemma, $\forall e \ : \ n_e \in \mathbb{N}$ as $c(e)$ is an integer power of $p$ for all edges in $E$. The graph $G^*$ of the network $\phi^*$ is constructed as follows. Starting with $G^* = (V, \emptyset)$, for each edge $e = (u, v) \in E$ we insert $n_e$ parallel edges connecting $u$ to $v$ in $G^*$, to form the edge set $E^*$. Additionally, we define the bond dimensions function of the network $\phi^*$ to assign the value of $p$ to all of the added edges, i.e. $\forall e^* \in E^* \ : \ c^*(e^*) = p$. In fig. 13 an example for such a construction of $\phi^*$ is shown for some $N = 8$ ConvAC TN.

In the paragraphs below, we derive eq. 25 which shows that an analysis of $\phi^*$ suffices for our purposes. This result is intuitive in some sense, as the construction of $\phi^*$ keeps intact the architecture of the network and the distribution of the degrees of freedom to some extent. As it is the key to our proof, we formulate this argument hereinafter.

As each edge $e \in E$ was translated into $n_e$ edges in $E^*$, there are $N^* := \log_p(\dim(\mathcal{H}^A) \cdot \dim(\mathcal{H}^B))$ external edges in $\phi^*$. Let $\mathcal{A}^*$ be the order $N^*$ tensor obtained by the contraction of the TN $\phi^*$. We denote by $(A^*, B^*)$ the partition of $[N^*]$ which corresponds to the partition $(A, B)$ of $[N]$. This means that an index number in $\mathcal{A}^*$ corresponding to an edge $e^* \in E^*$ would be in $A^*$ (resp. $B^*$) if the edge $e \in E$ from which it originated corresponded to an index number in $\mathcal{A}$ that was in $A$ (resp. $B$). This is easily understood pictorially, see fig. 13. Accordingly denote the corresponding partition of the degree 1 vertices in $G^*$ by $(V^{A*}, V^{B*})$. We will now show that the rank of the matricization of $\mathcal{A}$ w.r.t. the partition $(A, B)$ is equal to the rank of the matricization of $\mathcal{A}^*$ w.r.t. the partition $(A^*, B^*)$.

We denote by $\tau_v$ the tensors corresponding to a vertex $v \in V$ in the network $\phi$, and by $\tau_v^*$ the tensors corresponding to the same vertex $v$ in the network $\phi^*$. Let $z$ be the order of $\tau_v$, and denote the set of edges in $E$ incident to $v$ by $\{e_{k_1}, ..., e_{k_z}\}$ where $k_1, ..., k_z$ are the corresponding indices. For every index $k_j$, $j \in [z]$, let $K^{*j} = \{k_1^{*j}, ..., k_{n_{e_{k_j}}}^{*j}\}$ be the indices corresponding to the edges which were added to $\phi^*$ in the location of $e_{k_j}$ in $\phi$. According to the construction above, there is a one-to-one correspondence between the elements in $K^{*j}$ and $k_j$, that can be written as:

$$k_j = h(K^{*j}) := 1 + \sum_{t=1}^{n_{e_{k_j}}} p^{t-1}(k_t^{*j} - 1), \tag{23}$$

where $h \ : \ [p]^{\otimes n_{e_{k_j}}} \to [p^{n_{e_{k_j}}}]$. Thus, if one has the entries of the tensors in $\phi^*$, the following assignment to the entries of the tensors in $\phi$:

$$(\tau_v)_{k_1 ... k_z} = (\tau_v^*)_{h(K^{*1}) ... h(K^{*z})} \tag{24}$$

would ensure :

$$\mathrm{rank}(\llbracket \mathcal{A} \rrbracket_{A,B}) = \mathrm{rank}(\llbracket \mathcal{A}^* \rrbracket_{A^*, B^*}). \tag{25}$$

Effectively, we have shown that the claim to be proved regarding $\operatorname{rank}(\llbracket \mathcal{A} \rrbracket_{A,B})$ can be equivalently proved for $\operatorname{rank}(\llbracket \mathcal{A}^* \rrbracket_{A^*,B^*})$.

*The Form of the $\delta$ Tensor in $\phi^*$:*

It is worthwhile to elaborate on the form of a tensor in $\phi^*$ which corresponds to an order 3 $\delta$ tensor in $\phi$. We denote by $\tau_\delta^v = \delta$ a $\delta$ tensor in $\phi$, and by $\tau_\delta^{*v}$ the corresponding tensor in $\phi^*$. Fig. 14(a) shows an example for a transformed tensor in $\phi^*$ that originated in an order 3 $\delta$ tensor in $\phi$, all edges of which uphold $n_e = 2$. From eqs. 23 and 24, and from the form of the $\delta$ tensor given in eq. 12, it is evident that in this case an entry is non-zero in $\tau_\delta^{*v}$ only when $k_1^{*1} = k_1^{*2} = k_1^{*3}$ *and* $k_2^{*1} = k_2^{*2} = k_2^{*3}$. In the general case, the condition for an entry of 1 in $\tau_\delta^{*v}$ is:

$$\forall t \in [n_e]: \ k_t^{*1} = k_t^{*2} = k_t^{*3}, \tag{26}$$

where $n_e = \log_p c(e)$ for any edge $e$ incident to $v$ in $G$. Hence, a tensor $\tau_\delta^{*v}$ in $\phi^*$ which corresponds to a $\delta$ tensor in $\phi$ can be written as:

$$\tau_\delta^{*v} = \delta_{k_1^{*1} k_1^{*2} k_1^{*3}} \delta_{k_2^{*1} k_2^{*2} k_2^{*3}} \ldots \delta_{k_{n_e}^{*1} k_{n_e}^{*2} k_{n_e}^{*3}}. \tag{27}$$

*$\delta$ Restricted Edge Disjoint Paths*

Consider an edge-cut set in $G$ that achieves the minimal multiplicative weight over all cuts w.r.t the partition $(V^A, V^B)$ in the graph $G$: $C_{min} \in \operatorname{argmin}_C W_C$, and consider the corresponding edge-cut set $C_{min}^*$ in $G^*$ s.t. for each edge $e \in C_{min}$, the $n_e$ edges constructed from it are in $C_{min}^*$. By the construction of $G^*$, there are exactly $L := \log_p(\min_C W_C)$ edges in $C_{min}^*$ and their multiplicative weight upholds $W_{C_{min}^*} = W_{C_{min}} = p^L$.

A search for a minimal multiplicative cut, can be generally viewed as a classical min-cut problem when defining a maximum capacity for each edge that is a logarithm of its bond dimension. Then, a min-cut/max-flow value can be obtained classically in a graph with additive capacities and a final exponentiation of the result provides the minimal multiplicative value of the min-cut. Since all of the bond dimensions in $\phi^*$ are equal to $p$, such a process results in a network with all of its edges assigned capacity 1. From the application of theorem 3 on such a graph, it follows that the maximal number of *edge disjoint paths* between $V^{A*}$ and $V^{B*}$ in the graph $G^*$, which are paths between $V^{A*}$ and $V^{B*}$ that are allowed to share vertices but are not allowed to share edges, is equal to the cardinality of the minimum edge-cut set $C_{min}^*$. In our case, this number is $L$, as argued above. Denote these edge disjoint paths by $q_1, \ldots, q_L$.

In accordance with the form of $\tau_\delta^{*v}$, the tensors in $\phi^*$ corresponding to $\delta$ tensors in $\phi$ given in eq. 27, we introduce the concept of $\delta$ *restricted edge disjoint paths* between $V^{A*}$ and $V^{B*}$ in the graph $G^*$, which besides being allowed to share vertices but not to share edges, uphold the following restriction. For every $\delta$ tensor $\tau_\delta^v$ of order 3 in the graph $G$, with $e \in E$ a representative edge incident to $v$ in $G$, a maximum of $n_e$ such paths can pass through $v$ in $G^*$, each assigned with a different number $t \in [n_e]$. The paths uphold that when passing through $v$ in $G^*$ each path enters through an edge with index $k_{t_{in}}^{*j_{in}}$ and leaves through an edge with index $k_{t_{out}}^{*j_{out}}$ only if $j_{in} \neq j_{out} : j_{in}, j_{out} \in [3]$ and $t_{in} = t_{out} = t$, where no two paths can have the same $t$. This restriction imposed on the indices of $\tau_\delta^{*v}$ in $\phi^*$, to be called hereinafter the $\delta$ restriction, is easily understood pictorially, e.g. in fig. 14(a) the paths crossing the $\tau_\delta^{*v}$ tensor must only contain edges of the same color in order to uphold the $\delta$ restriction.

We set out to show, that for the network in question one can choose the $L$ edge disjoint paths to uphold the $\delta$ restriction. Then, a weight assignment compliant with the $\delta$ tensors in the network can be guaranteed to uphold the requirements of the lemma, despite the fact that most of the entries in the $\delta$ tensors are equal to zero.

Denote the set of $n_e$ edges in $G^*$ that originated from a certain edge $e$ in $G$, by $X_e^* \subset E^*$. We first show that one can choose the $L$ edge disjoint paths s.t. in a flow directed from $V^{A*}$ to $V^{B*}$ w.l.o.g, there is no set of edges $X_e^*$ that corresponds to any $e \in E$ for which two edges $e_i^*, e_j^* \in X_e^* \subset E^*$ belong to paths $q_i, q_j$ which flow in opposite directions. Fig. 14(b) clarifies this claim.

We observe the classical max-flow in the graph $G$, *i.e.* when assigning a maximum capacity for each edge $e$ that is equal to $n_e := \log_p c(e)$, a maximum flow of $L$ is possible between $V^A$ and $V^B$ in $G$. Observing the paths in $G$ that flow w.l.o.g. from $V^A$ to $V^B$, together they can transfer a maximum capacity of $L$. Note that in $G$, these paths most certainly do not need to be edge disjoint paths. We argue that one can choose such paths from $V^A$ to $V^B$ in $G$ such that on each edge $e$ there is an integer capacity transferred. The existence of such paths in $G$ follows directly from the integral flow theorem (Dantzig and Fulkerson (1956)), which states that if each edge has integral capacity, then there exists such an integral maximal flow. Note, that these paths must also uphold the basic rule that the sum of capacities transferred on a certain edge $e \in E$, even if this is performed via several paths, is less than the edge maximum capacity $n_e$.

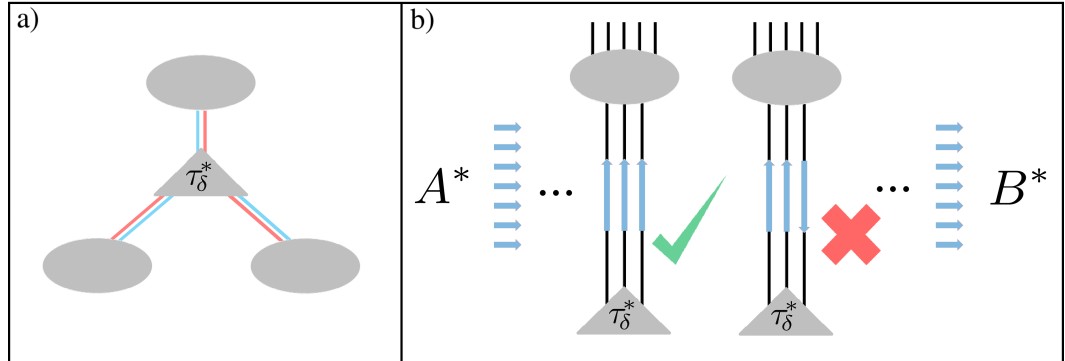

Figure 14: (a) An example for the tensor in $\phi^*$ which corresponds to a $\delta$ tensor $\tau_\delta^v \in \mathbb{R}^{p^2 \times p^2 \times p^2}$ in $\phi$. According to the construction of $\phi^*$ presented in fig. 13, each edge is split into $n_e = 2$ edges of bond dimension $p$. The $\delta$ tensor structure in $\phi$ translates into this $\tau_\delta^{*v}$ tensor holding a non-zero entry only when the indices corresponding to all of the edges that are marked by the same color are equal to each other (eq. 27). Additionally, paths crossing this $\tau_\delta^{*v}$ tensor must only contain edges of the same color in order to be called $\delta$ restricted edge disjoint paths. (b) There are $L$ guaranteed edge disjoint paths between $V^{A*}$ and $V^{B*}$. In a flow directed from $V^{A*}$ to $V^{B*}$ (w.l.o.g), we argue that one can choose these paths such that they have the same flow direction in all edges in $\phi^*$ that originate from a certain edge in $\phi$.

One can now construct $L$ paths in $G^*$ in a recursive manner, if a maximum additive capacity for each edge $e* \in E^*$ is similarly defined to be $\log_p c^*(e^*) = \log_p p := 1$. Starting with a single quanta of flow along some path in $G$, construct a single path in the corresponding position in $G^*$. Each edge that is part of this path in $G^*$ will transfer exactly one quanta of flow, as that is their maximum capacity that is chosen to be saturated in order to transfer the same amount of capacity that is transferred in $G$. Now, remove the full edges in $G^*$ and reduce the capacities of all edges along the original path in $G$ by one. Repeating this process until a capacity of $L$ is transferred in both graphs, since $n_e$ is the number of new edges added to $G^*$ in the place of each edge $e$, and it is also an upper bound on the *integer* capacity this path transfers in $G$, it follows that in $G^*$ one finds $L$ paths between $V^{A*}$ and $V^{B*}$ that correspond exactly to the paths transferring integer capacity in $G$ guaranteed by integral flow theorem. These paths in $G^*$ are edge disjoint since the edges of each path were removed from the graph when constructed. Choosing precisely these edge disjoint paths in $G^*$, one is guaranteed that the flow from $V^{A*}$ to $V^{B*}$ in all of the edges in $X_e^*$ that belong to these paths would be in the same direction, as they originated in the same edge $e$ in $G$ that had a flow in that single specific direction from $A$ to $B$. Pictorially, since the different edges in $X_e^*$ all originate from one single edge that obviously cannot have two opposite directions of net flow, they can all be chosen to transfer flow in the same direction.

Observing an order 3 $\delta$ tensor $\tau_\delta^v$ in $\phi$, denote the three edges incident to $v$ in $G$ by $e_1, e_2, e_3 \in E$, and denote $n_e := n_{e_1} = n_{e_2} = n_{e_3}$. Now that we have asserted that all of the $L$ edge disjoint paths in $G^*$ may uphold the above condition, we choose the paths as such, *i.e.* under this choice all of the edges in each respective set (namely $X_{e_1}^*, X_{e_2}^*$ or $X_{e_3}^*$) pass flow from $V^{A*}$ to $V^{B*}$ in the same direction. In this case, a maximum of $n_e$ paths can pass through the delta tensor. This can be easily understood by the following argument. Denote a set $X_{e_i}^*$ by 'I' if the paths passing through its edges are incoming to the $\delta$ tensor in a flow from $V^{A*}$ to $V^{B*}$, and by 'O' if they are outgoing from the $\delta$ tensor in such a flow. W.l.o.g. we assume that $X_{e_1}^*, X_{e_2}^*$ are denoted by 'I' and $X_{e_3}^*$ is denoted by 'O'. in this case, only $n_e$ such edge disjoint paths can flow *out* of the $\delta$ tensor. In the opposite case, where two groups of edges out of the three are denoted by 'O' and only one group is denoted by 'I', only $n_e$ such edge disjoint paths can flow *into* the $\delta$ tensor. The contrary, *i.e.* if more than $n_e$ such paths were to cross the $\delta$ tensor, would imply a cross flow of edge disjoint paths in at least one of the sets $X_{e_1}^*, X_{e_2}^*, X_{e_3}^*$, in contradiction to this choice of paths.

This provides us with the ability to distribute the paths in the following manner, that upholds the $\delta$ restriction described above. Assume w.l.o.g that $X_{e_1}^*$ is the set for which the most edges are in the chosen edge disjoint paths. Denote by $q_1, ..., q_{N_2}$ the paths that include edges in $X_{e_1}^*$ and $X_{e_2}^*$, and by $q_{N_2+1}, ..., q_{N_2+N_3}$ the paths that include edges in $X_{e_1}^*$ and $X_{e_3}^*$. Finally, assign the index $t$ to the path $q_t$. From the statement above, it is guaranteed that $N_2 + N_3 \leq n_e$. Therefore, this choice of paths is guaranteed to uphold the delta restriction defined above, which states that each path must receive a *different* value $t \in [n_e]$. Specifically, this implies that the maximal number of $\delta$ restricted edge disjoint paths between $V^{A*}$ and $V^{B*}$ in the graph $G^*$ is $L$.

*The Assignment of Weights:*

We give below explicit tensor assignments for all the tensors in $\phi^*$ so that $\text{rank}(\llbracket \mathcal{A}^* \rrbracket_{A^*,B^*}) = \min W_C$, which in accordance with eq. 25 implies that $\text{rank}(\llbracket \mathcal{A} \rrbracket_{A,B})$ upholds this relation. Together with the translation from $\phi^*$ to $\phi$ given in eq. 24, this will constitute a specific example of an overall assignment to the TN representing the ConvAC which achieves the lower bound shown in this lemma.

Observing the form of $\phi^*$, an example for which is shown in fig. 13, we see that it is composed of tensors that correspond to $\delta$ tensors in $\phi$, denoted by $\tau_\delta^{*v}$, and of tensors that correspond to weight matrices in $\phi$, denoted by $A^{*(l,j)}$. Recalling that the entries of the former are dictated by construction and obey eq. 27, we are left with providing in assignment of all the tensors $A^{*(l,j)}$. The weight matrices in a ConvAC TN uphold $A^{(l,j)} \in \mathbb{R}^{r_l \times r_{l-1}}$, thus the corresponding tensors $A^{*(l,j)}$ are of order $\log_p(r_l \cdot r_{l-1})$ by construction, with $\log_p r_l$ edges directed upwards in the network and $\log_p r_{l-1}$ edges directed downwards. For clarity, we omit the superscript from $A^{*(l,j)}$ and write the indices of such a weights tensor as:

$$A^{k_1 \dots k_{\log_p r_l}}_{k_{\log_p r_l + 1} \dots k_{\log_p (r_l \cdot r_{l-1})}}. \tag{28}$$

We choose $L$ paths between $V^{A*}$ and $V^{B*}$ in the graph $G^*$ which are $\delta$ restricted edge disjoint paths, denoted by $q_1, ..., q_L$. We are guaranteed to have this amount of $\delta$ restricted edge disjoint paths by the arguments made in the previous subsection. For any weights tensor $A^{*(l,j)}$ in $\phi^*$, let $n \in [\min(L, \lfloor \frac{1}{2} \log_p(r_l \cdot r_{l-1}) \rfloor)] \vee \{0\}$ be the number of $\delta$ restricted edge disjoint paths crossing it.

Let $\{g_{1\alpha}, g_{1\beta}, ..., g_{n\alpha}, g_{n\beta}\}$ with $g_{ix} \in [\log_p(r_l \cdot r_{l-1})]$, $i \in [n]$, $x \in \{\alpha, \beta\}$ be the numbers representing indices of $A^{*(l,j)}$ which correspond to edges that belong to any path $q_1, \dots, q_L$, *i.e.* the set of such indices is $G := \{k_{g_{1\alpha}}, k_{g_{1\beta}}, ..., k_{g_{n\alpha}}, k_{g_{n\beta}}\}$. Denoting $\bar{n} := \log_p(r_l \cdot r_{l-1}) - 2n$, let $\{f_1, ..., f_{\bar{n}}\}$ with $f_i \in [\log_p(r_l \cdot r_{l-1})]$, $i \in [\bar{n}]$ be the numbers representing the remaining indices of $A^{*(l,j)}$, *i.e.* the indices which correspond to edges that are not on any path $q_j$, $j \in [L]$. The set of such indices is $F := \{k_{f_1}, ..., k_{f_{\bar{n}}}\}$. Since each edge in the graph connects a weights matrix with a $\delta$ tensor (see fig. 13), we may identify each such $k \in F$ with an index $k_t^{*j}$, $j \in [3], t \in [n_e]$ of the adjacent delta tensor. We accordingly define:

$$\eta(k) = \begin{cases} \delta_{k1} & \forall j, \ k_t^{*j} \text{ does not belong to any path } q_1, \dots, q_L \\ 1 & else \end{cases} \tag{29}$$

Finally, the assignment of the entries of $A^{*(l,j)}$ is given by:

$$A^{k_1 \dots k_{\log_p r_l}}_{k_{\log_p r_l + 1} \dots k_{\log_p (r_l \cdot r_{l-1})}} = \delta_{k_{g_{1\alpha}} k_{g_{1\beta}}} \cdots \delta_{k_{g_{n\alpha}} k_{g_{n\beta}}} \eta(k_{f_1}) \cdots \eta(k_{f_{\bar{n}}}). \tag{30}$$

Effectively, the assignment in eq. 30 for the weights tensors ensures that their indices which correspond to two edges that are adjacent in one of the paths $q_i$, must be equal in order for the term not to vanish in the contraction of the entire TN $\phi$. Since the paths $q_i$ are $\delta$ restricted, the $\tau_\delta^{*v}$ tensors in $\phi^*$ which corresponds to a $\delta$ tensor in $\phi$ are also able uphold this rule a priori. By this assignment, in accordance with the form of $\tau_\delta^{*v}$ given in eq. 27, the indices corresponding to all of the edges in a path $q_i$ are in fact enforced to receive the same value, ranging in $[p]$, in order for the term not to vanish in the contraction. An additional result of this assignment, is that all of the indices in the network which correspond to edges that do not belong to any path $q_i$, must be equal to 1 in order for the term not to vanish (*i.e.* when they receive values of $2, ..., p$ the term vanishes upon contraction).

According to the rules of TNs introduced in appendix C, the overall tensor $\mathcal{A}^*$ represented by the network $\phi^*$ is calculated by a global contraction which is a summation over all of the internal indices. Under the assignment in eq. 30, upon a simple rearrangement of the tensor modes s.t. indices corresponding to $A^*$ are on the left, indices corresponding to $B^*$ are on the right and the indices corresponding to paths are first in their respective regions,[11] the only non-zero entries of $\mathcal{A}^*$ are:

$$\mathcal{A}^*_{d_{q_1} \dots d_{q_L}} \overbrace{1 \ \dots \ 1}^{|A^*| - L} d_{q_1} \dots d_{q_L} \overbrace{1 \ \dots \ 1}^{|B^*| - L} = 1, \tag{31}$$

where $d_{q_1}, ..., d_{q_L} \in [p]$ are the indices corresponding to the paths $q_1, ..., q_L$, respectively. Upon matricization of $\mathcal{A}^*$ w.r.t. the partition $(A^*, B^*)$, it is clear that a matrix of rank $p^L = \min_C W_C$ with $I_{p^L \times p^L}$ on its upper left block and zeros otherwise is received, and the lemma follows. $\qquad\square$

---

[11]This does not affect the rank of matricization as it is still performed w.r.t. the partition $(A^*, B^*)$, and switching rows or columns leaves the rank unchanged.

With all the preliminaries in place, the proof of theorem 2 readily follows:

*Proof.* (of theorem 2)

For a specific $p$, consider the network $\phi^p$ such as defined in theorem 2, *i.e.* a TN with the same connectivity as $\phi$, where all of the bond dimensions are modified to be equal the closest power of $p$ to their value in $\phi$ from below. Let $(\mathcal{A}^p)^y$ be the weights tensor represented by $\phi^p$. According to lemma 2, such a network upholds that the rank of the matricization $[\![(\mathcal{A}^p)^y]\!]_{A,B}$ is at least: $\min_C W_C^p$ almost always. According to lemma 1, a specific assignment for the weights of the tensors in $\phi^p$ that achieves this value suffices to imply that $[\![\mathcal{A}^y]\!]_{A,B}$ is at least: $\min_C W_C^p$ almost always, e.g. the assignment given in lemma 2. Specifically, this holds for $\phi^p$ with $p \in \operatorname{argmax}_p \min_C W_C^p$, satisfying the theorem. $\qquad\square$

## F  REPRODUCING DEPTH EFFICIENCY

The exponential depth efficiency result shown in Cohen et al. (2016b), can be straightforwardly reproduced by similar graph-theoretic considerations. We show below an upper bound on the rank of matricization of the conv-weights tensor for a case of a general pooling window. The bound implies that any amount of edges in a cut that are connected to the same $\delta$ tensor will contribute their bond dimension only once to the multiplicative weight of this cut, thus effectively reducing the upper bound when many cut edges belong to the same $\delta$ tensor. This does not affect our analysis of the deep network above, as the $\delta$ tensors in that network are only three legged (see fig. 8). Therefore, in the above analyzed deep ConvAC, a cut containing more than one $\delta$ tensor leg can be replaced by an equivalent cut containing only one leg of that $\delta$ tensor, and the value of $\min_C W_C$ is unchanged.

Formally, in order to apply similar considerations to the ConvAC with general sized pooling windows, such as the one presented in fig. 6, one must consider more closely the restrictions imposed by the $\delta$ tensors. To this end, we define the object underlying a ConvAC-weights TN with general sized pooling windows $\phi$ to be composed of the following three:

- An undirected graph $G(V, E)$, with a set of vertices $V$ and a set of edges $E$. The set of nodes is divided into two subsets $V = V^{\text{tn}} \uplus V^{\text{inputs}}$, where $V^{\text{inputs}}$ are the $N$ degree-1 virtual vertices and $V^{\text{tn}}$ corresponds to tensors of the TN.
- A function $f : E \to [b + N]$, where $b$ is the number of $\delta$ tensors in the network. If we label each $\delta$ tensor in the network by a number $i \in [b]$, this function upholds $f(e) = i$ for $e \in E$ that is incident to a vertex which represents the $i^{th}$ delta tensor in the ConvAC TN. For each edge $e \in E$ incident to a degree 1 vertex, this function assigns a different number $f(e) = i$ for $i \in b + [N]$. Such an edge is an external edge in the ConvAC TN, which according to the construction presented in appendix D is the only type of edge not incident to any $\delta$ tensor. In words, the function $f$ divides al the edges in $E$ into $b + N$ groups, where edges are in the same group if they are incident to the same vertex which represents a certain $\delta$ tensor in the network.
- A function $c : [b + N] \to \mathbb{N}$, associating a bond dimension $r \in \mathbb{N}$ with each different group of edges defined by the set: $E_i = \{e \in E : f(e) = i\}$.

Observing an edge-cut set with respect to the partition $(A, B)$ and the corresponding set $G^C = \{f(e) : e \in C\}$. We denote the elements of $G^C$ by $g_i^C, i \in [|G_C|]$. These elements represent the different groups that the edges in $C$ belong to (by the definition of $f$, edges incident to the same delta tensor belong to the same group). We define the modified weight of such an edge-cut set $C$ as:

$$\tilde{W}_C = \prod_{i=1}^{|G_C|} c(g_i^C). \tag{32}$$

The weight definition given in eq. 32 can be intuitively viewed as a simple multiplication of the bond dimensions of all the edges in a cut, with a single restriction: the bond dimension of edges in the cut which are connected to a certain $\delta$ tensor, will only be multiplied once (such edges have equal bond dimensions by definition, see eq. 12). An example of this modified weight can be seen in fig. 15, where the replacement of a general tensor by a $\delta$ tensor results in a reduction in the minimal cut, due to the rules defined above. In the following claim, we provide an upper bound on the ability of a ConvAC with a general pooling window to model correlations of its inputs, as measured by the Schmidt entanglement measure (see section 4).

**Claim 3.** *Let $(A, B)$ be a partition of $[N]$, and $[\![\mathcal{A}^y]\!]_{A,B}$ be the matricization w.r.t. $(A, B)$ of a conv-weights tensor $\mathcal{A}^y$ realized by a ConvAC depicted in fig. 6 with a general pooling window. Let $G(V, E, f, c)$ represent the TN $\phi$ corresponding to the ConvAC network, and let $(V^A, V^B)$ be the vertices partition of $V^{inputs}$ in $G$ corresponding to $(A, B)$. Then, the rank of the matricization $[\![\mathcal{A}^y]\!]_{A,B}$ is no greater than: $\min_C \tilde{W}_C$, where $C$ represents a cut w.r.t $(V^A, V^B)$ and $\tilde{W}_C$ is the modified multiplicative weight defined by eq. 32.*

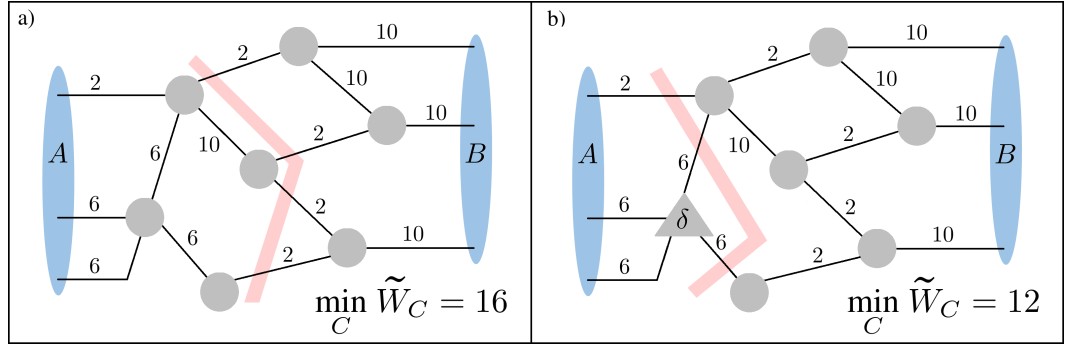

Figure 15: An example for the effect that a $\delta$ tensor has on the upper bound on the rank of the matricization of the overall tensor represented by a TN. $\min_C \tilde{W}_C$ is defined in eq. 32 and shown in claim 3 to be the upper bound on the rank of the matricization of the conv-weights tensor of a ConvAC represented by a TN. In this example, the upper bound is reduced upon changing a single general tensor in the TN shown in (a) (identical to fig. 11), whose entries are free to be equal any value, with a $\delta$ tensor in the TN shown in (b) which obeys the constraint given in eq. 12. The centrality of the $\delta$ tensor in the TN compliant with a shallow ConvAC (that is depicted in fig. 7, is in effect the element which limits the expressiveness of the shallow network, as is discussed in appendix F.

Having seen the proof of the claim 1 above and its accompanying graphics, the proof of the upper bound presented in claim 3 can be readily attained. The only difference between the two lies in the introduction of the $\delta$ tensors to the network, which allows us to derive the tighter lower bound shown in claim 3.

*Proof.* (of claim 3)

The modification to the above proof of claim 1 focuses on the coalescence of the cut indices $I_C$ into a single index $m \in [\prod_{i=1}^{|C|} c_{k_i}]$. Assume that any two indices of multiplicands in this product, denoted by $k_i$ and $k_j$, are connected to the same $\delta$ tensor that has some bond dimension $q := c_{k_i} = c_{k_j}$. Upon contraction of the TN in fig. 12(b), the cut indices are internal indices that are to be summed upon. However, whenever $k_i \in [q]$ and $k_j \in [q]$ are different, by the constraint imposed in the $\delta$ tensor definition (eq. 12), the entire term vanishes and there is no contribution to the final value of $\mathcal{A}_{d_1 \ldots d_N}$ calculated by this contraction. Thus, $k_i$, $k_j$ and any other index connected to the same $\delta$ tensor can be replaced by a representative index $k^\alpha \in [q]$ whenever they appear in the summation. $\alpha \in G^C$ upholding $c(\alpha) = q$, is the group index of this $\delta$ tensor, given by $\alpha = f(e_{k_i}) = f(e_{k_j})$ with $e_{k_i}$ and $e_{k_j}$ the edges corresponding to the indices $k_i$ and $k_j$ in the network. Thus, the single index $m$ achieved by coalescing all of the cut indices can be defined in the range $m \in [\tilde{W}_C]$, with $\tilde{W}_C$ defined by eq. 32 upholding $\tilde{W}_C \leq \prod_{i=1}^{|C|} c_{k_i}$, where the equality is satisfied when no two edges in the cut are incident to the same $\delta$ tensor. Finally, the matricization $[\![\mathcal{A}]\!]_{A,B}$ can be written as a multiplication of two matrices as portrayed in fig. 12(c):

$$([\![\mathcal{A}]\!]_{A,B})_{lr} = \sum_{m=1}^{\tilde{W}_C} ([\![\mathcal{X}]\!]_{A,C})_{lm} ([\![\mathcal{Y}]\!]_{C,B})_{mr}, \tag{33}$$

$l \in [\dim(\mathcal{H}^A)]$, $r \in [\dim(\mathcal{H}^B)]$. Recalling that as in the proof of claim 1 the edge-cut set may include the external edges, we attain:

$$\text{rank}([\![\mathcal{A}]\!]_{A,B}) \leq \min_C \tilde{W}_C. \tag{34}$$

$\square$

Observing fig. 7 which shows the TN corresponding to the shallow ConvAC architecture, the central positioning of a single $\delta$ tensor implies that under *any* partition of the inputs $(A, B)$ s.t. $|A| = |B| = N/2$, the minimal cut will obey $W_C^{\min} = \min(M^{N/2}, k)$. Thus, in order to reach the exponential in $N$ measure of entanglement w.r.t. the interleaved partition that was obtained in section 5 for the deep network, the number of channels in the single hidden layer of the shallow network $k$, must grow exponentially with $N$. Therefore, one must exponentially enlarge the size of the shallow network in order to achieve the expressiveness that a polynomially sized deep network achieves, and an exponential depth efficiency is demonstrated.

# G   UPPER BOUND DEVIATIONS SIMULATION

In this section, we describe simulations performed on an $N = 16$ deep ConvAC TN (with pooling windows of size 2), which are aimed at quantifying the prevalence of deviations from the upper bound on the ranks of the matricization of conv-weights tensor presented in claim 1. In section E.2 we proved theorem 2, showing in effect that this upper bound is tight when all of the channel numbers are powers of some integer $p$, and guaranteeing a positive result in all cases. However, for the general setting of channel numbers there is no theoretical guarantee that the upper bound is tight. Indeed, Cui et al. (2016) show a counter example where the matricization rank is effectively lower the minimal multiplicative cut for a general TN (that has no $\delta$ tensors such as in the ConvAC TN). There is no reason to believe that the upper bound is tight for the TN representing a ConvAC for a general setting of channel numbers, and indeed our simulations below show deviations from it. However, as is indicated below such deviations are negligible in prevalence and low in value. A theoretical formulation of this is left for future work.

The experiments were performed in matlab, and tensor contractions were computed using a function introduced by Pfeifer et al. (2014). An $N = 16$ with $M = 2$ ConvAC TN was constructed (see figs. 8 and 9), with the entries of the weights matrices randomized according to a normal distribution. The bond dimensions of layers 0 through 3 were drawn from the set of the first 6 prime numbers: $[2, 3, 5, 7, 11, 13]$, to a total of 360 different arrangements of bond dimensions. This was done in order to resemble a situation as distinct as possible from the case where all of the bond dimensions are powers of the same integer $p$, for which the tightness of the upper bound is guaranteed by theorem 2. Per bond dimension arrangement, all of the $\frac{1}{2} \cdot \binom{16}{8} = 6435$ different partitions were checked, for a total of $360 \cdot 6435 = 2.3166 \cdot 10^6$ different configurations. As argued in section E.2, the logarithm of the upper bound on the rank of the conv-weights tensor matricization that is shown in claim 1, is actually the max-flow in a network with the same connectivity that has edge capacities which are equal to the logarithm of the respective bond dimensions. Therefore, a configuration for which the rank of matricization is equal to the exponentiation of the max-flow through such a corresponding network, effectively reaches the upper bound. We calculated the max-flow independently for each configuration using the Ford-Fulkerson algorithm (Ford and Fulkerson (1956)), and set out to search for deviations from such an equivalence.

The results of the above described simulations are as follows. Only 1300 configurations, which constitute a negligible fraction of the 2.3166 million configurations that were checked, failed to reach the upper bound and uphold the min-cut max-flow equivalence described above. Moreover, in those rare occasions that a deviation occurred, the percentage of deviations from the upper bound did not exceed 10% of the value of the upper bound. This check was performed on a bond setting that is furthest away from all channel numbers being powers of the same integer, yet the tightness of the upper bound emerges as quite robust, justifying experimentally our general view of the minimal weight over all cuts in the network, $\min_C W_C$, as the effective indication for the matricization rank of the conv-weights tensor w.r.t. the partition of interest. A caveat to be stated with this conclusion is that we checked only up to $N = 16$, and the discrepancies that were revealed here might become more substantial for larger networks. As mentioned above, this is left for future theoretical analysis, however the lower bound shown in theorem 2 guarantees a positive result regarding the rank of the matricization of the conv-weights tensor in all of the cases.

