# OpenReview forum: "Deep Learning and Quantum Entanglement: Fundamental Connections with Implications to Network Design"
_ICLR.cc/2018/Conference — Accept (Poster)_

### Official Review · AnonReviewer3 · 2017-12-07
**Interesting theoretical connections**

**Rating:** 7
**Confidence:** 3

**Review:**

The paper makes a striking connection between two apparently unrelated problems: the problem of designing neural networks to handle a certain type of correlation and the problem of designing a structure to represent wave-function with quantum entanglement. In the wave-function context, the Schmidt decomposition of the wave function is an inner product of tensors. Thus, the mathematical glue connecting the neural networks and quantum entanglement is shown to be tensor networks, which can represent higher order tensors through inner product of lower-order tensors.

The main technical contribution in the paper is to map convolutional networks with product pooling function (called ConvACs) to a tensor network. Given this mapping, the authors exploit results in tensor networks (in particular the quantum max-flow min-cut theorem) to calculate the rank of the matricized tensor between a pair of vertex sets using the (appropriately defined) min-cut.

The connection has potential to yield fruitful new results, however, the potential is not manifested (yet) in the paper. The main application in deep convolutional networks proposed by the paper is to model how much correlation between certain partition of input variables can be captured by a given convolutional network design. However, it is unclear how to use Theorem 1 to design neural networks that capture a certain correlation.

A simple example is given in the experiment where the wider layers can be either early in the the neural network or at the later stages; demonstrating that one does better than the other in a certain regime. It seems that there is an obvious intuition that explains this phenomenon: wider base networks with large filters are better suited to the global task and narrow base networks that have more parameters later down have more local early filters suited to the local task. The experiments do not quite reveal the power of the proposed approach, and it is unclear how, if at all, the proposed approach can be applied to more complicated networks.

In summary, this paper is of high theoretical interest and has potential for future applications.

---

> ### Public Comment · (anonymous) · 2017-12-12
> **Authors' response to reviewer 3**
>
> We thank the reviewer for the feedback and support.
>
> The example we give for practical guidelines relates to layers widths - wider base networks, with more
> parameters in shallower layers, are better fit to model local input correlations and vice versa. As another
> example, Theorem 1 implies that the contiguous pooling scheme commonly used in deep networks is
> more appropriate when short-range correlations are present in the data, and that a different pooling
> scheme which merges symmetric activations is preferable when long-range correlations are present
> (this is pointed out and verified experimentally by [1], which uses other methods).
> Conclusions obtained by relying on the min-cut analysis in Theorem 1 indeed *exactly* hold only for
> the network on which it was proven (ConvAC), however the experiments performed in our paper and in
> [1] provide evidence that such conclusions extend to other commonly used architectures.
>
> Reference
> --------------------------
> [1] Nadav Cohen and Amnon Shashua. Inductive bias of deep convolutional networks through pooling
> geometry. In 5th International Conference on Learning Representations (ICLR), 2017

---

### Official Review · AnonReviewer1 · 2017-12-08
**Good contribution, accept**

**Rating:** 8
**Confidence:** 5

**Review:**

The authors try to bring in two seemingly different areas and try
to leverage the results in one for another.
First authors show that the equivalence of the function realized(in
tensor form, given in earlier work) by a ConvAC and
the function used to model n-body quantum system. After establishing
the equivalence of two, the authors argue that
quantum entanglement measures used to measure correlations in n-body
quantum systems can be used as an expressive measure
(how much correlation in input they can handle) of the function
realized by a ConvAC. Separation Rank analysis, which was done
earlier, becomes a special case. As the functional equivalence is
established, authors adopt Tensor Network framework,
to analyze the properties of the ConvAC. The main result being able
to quantify the expressiveness to some extend to the min
cut of the underlying Tensor Network graph corresponding to ConvAC.
This is further used to argue about guide-lining the
width of various parts of ConvAC, if some prior correlation
structure is known about the input. This is also validated
experimentally.

Although I do not see major results at this moment, this work can be
of great significance. The attempt to bring in two areas
have to be appreciated. This work opens up a footing to do graph
theoretical analysis of deep learning architectures and from
the perspective of Quantum entanglement, this could lead to open up new directions.
The paper is lucidly written, comprehensively covering the
preliminaries. I thoroughly enjoyed reading it, and I think the
paper and the work would be of great contribution to the community.

(There are some typos  (preform --> perform ))

---

> ### Public Comment · (anonymous) · 2017-12-12
> **Authors' response to reviewer 1**
>
> We thank the reviewer for the supporting feedback!

---

### Official Review · AnonReviewer5 · 2017-12-11
**Interesting theoretical connections and practical implications**

**Rating:** 6
**Confidence:** 2

**Review:**

This paper draws an interesting connection between deep neural networks and theories of quantum entanglement. They leveraged the tool for analyzing quantum entanglement to deep neural networks, and proposed a graph theoretical analysis for neural networks. They demonstrated how their theory can help designing neural network architectures on the MNIST dataset.

I think the theoretical findings are novel and may contribute to the important problem on understanding neural networks theoretically. I am not familiar with the theory for quantum entanglement though.

---

> ### Public Comment · (anonymous) · 2017-12-12
> **Authors' response to reviewer 5**
>
> We thank the reviewer for the time and feedback.

---

### Official Review · AnonReviewer6 · 2017-12-15
**Beautiful story, good writing, weak proof between TN and ConvNet, poor experiment**

**Rating:** 6
**Confidence:** 4

**Review:**

The paper proposes a structural equivalence between the function realised by a convolutional arithmetic circuit (ConvAC) and a quantum many-body wave function, which facilitates the use of quantum entanglement measures as quantifiers of a deep network’s expressive ability to model correlations.

The work is definitely worthwhile digging deeper, bridging some gap and discussions between physics and deep learning. The ultimate goal for this work, if I understands correctly, is a provide a theoretical explanation to the design of deep neural architectures. The paper is well-written (above most submissions, top 10%) with clear clarity. However, removing all the fancy stuff and looking into the picture further, I have several major concerns.

+ Potential good research direction to connect physical sciences (via TN) to deep learning theories (via ConvAC).

- [Novelty is limited and proof is vague] The paper uses physical concepts to establish a "LAYER WIDTHS EFFECT ON THE EXPRESSIVENESS OF A DEEP NETWORK", the core theory (proposed method) part is Section 5 alone and the rest (Section 2,3,4) is for introductory purposes. Putting Theorem 1 in simple, deep learning based English, it says for a dataset with features of a size D, there exists a partition of length scale \epsilon < D, which is guaranteed to separate between different parts of a feature. Based on this, they give a rule of thumb to design the width (i.e., channel numbers) of layers in a deep neural network: (a) layer l = logD is more important than those of deeper layers; (b) among these deeper layers, deeper ones need to be wider, which is derived from the min-cut in the ConvAC TN case. How (a) is derived or implied from theorem 1?

It seems to me that the paper goes with a rigorous manner till the proof of theorem 1, with all the concepts and denotations well demonstrated. Suddenly when it comes to connecting the practical design of deep networks, the conclusion becomes qualitative without much explanation via figures or visualisation of the learned features to prove the effectiveness of the proposed scheme.

- [Experiments are super weak] The paper has a good motivation and a beautiful story and yet, the experiments are poor to verify them. The reason as to why authors use ConvAC is that it more resembles the tensor operations introduced in the paper. There is a sentence, "Importantly, through the concept of generalized tensor decompositions, a ConvAC can be transformed to a standard convolutional network with ReLU activation and average/max pooling", to tell the relation between ConvAC and traditional convolutions. The theory is based on the analysis of ConvAC, and all of a sudden the experiments are conducted on the traditional convolution. This is not rigorous and not professional for a "technically-sound" paper. How the generalized concepts of tensor decompositions can be applied from ConvAC to vanilla convolutions?

The experiments seem to extend the channel width of *all* layers in a hand-crafted manner (10, 4r, 4r, xxx). Based on the derived rule of thumb, the most important layer in MNIST should be layer 3 or 4 (log 10). Some simple ablative analysis should be:
(i) baseline: fix layer 3, add more layers thereafter in the network;
(ii) fix layer 3, reduce the channel numbers after layer 3.
The (ii) case should be at least comparable to (i) if theorem 1 is correct.

Moreover, to verify conclusion (b) which I mentioned earlier, does the current setting (10, 4r, 4r, xx) consider "deeper ones need to be wider"? What is the value of r? MNIST is a over-used dataset and quite small. I see the performance in Figure 4 (the only experiment result in the paper) just exceeds 90%. A simple trained NN (not CNN) could reach well 96% or so.

More ablative study (convAC or vanilla conv, other datasets, comparison components in width design, etc.) are seriously needed. Otherwise, it is just not convincing to me.

If the authors target on the network design in a more general manner (not just in layer width, but the design of number of filters, layers, etc.), there are already some neat work in this community and you should definitely compare them: e.g., Neural Architecture Search with Reinforcement Learning, ICLR 2017. I know the paper starts with building the connection from physics to deep learning and it is natural to solve the width design issue alone. This is not a major concern.

-------------
We see lots of fancy conceptions, trying to bind interdisciplinary subjects to help analyse deep learning theories over the last few years, especially in ICLR 2017. This year same thing happens. I am not degrading the motivation/intuition of the work; instead, I think it is pretty novel to explain the design of neural nets, by way of quantum physics. But the experiments and the conclusion derived from the analysis make the paper not solid to me and I am quite skeptical about its actual effectiveness.

---

> ### Author Response · Authors · 2017-12-18
> **Authors' response to reviewer 6**
>
> We thank the reviewer for the time and feedback; our response follows.
>
> - Theory:
>
> There seems to be a misunderstanding.
> The rephrasing of our results made by reviewer (text following "in simple, deep learning based English") is incorrect. Reviewer claims that the implications of theorem 1 are:
> (a) Width of layer log(D), where D is the typical size of a feature, is more important than those of deeper layers.
> (b) Among these less important deep layers, deeper ones should be wider.
> The actual implications of theorem 1 are as stated explicitly at the end of sec 5:
> (i) Width of layers *up to* log(D) are more important than those of deeper layers.
> (ii) Among the layers that *are important*, i.e. layers 1...log(D), deeper ones should be wider.  The widths of the less important layers, i.e. layers log(D)+1 and up, are irrelevant.
>
> Implications (i) and (ii) above are a straightforward consequence of theorem 1, which characterizes correlations in terms of min cuts in a TN graph. Specifically, with features of size D, a partition that splits a feature will lead to a min cut that necessarily does not include edges corresponding to layers log(D)+1 and up.  Edges corresponding to layers 1...log(D) on the other hand will be included, hence the lesser need to strengthen these edges by widening the layers.  We did not frame implications (i) and (ii) above as a theorem, as they are a direct consequence of theorem 1, and we prefer to keep the text fluent and compact. Having said that, we realize from the review that additional details may assist the reader, and so have added such to the end of sec 5.
>
> A final note on theory:
> Reviewer writes "the paper goes with a rigorous manner till the proof of Theorem 1".  We would like to stress in the most explicit manner that unlike the derivation of conclusions (i) and (ii) from theorem 1, the theorem itself is highly non-trivial.  It is stated formally in the body of the paper, and proven in full in the appendix (pages 17-27).
>
> - Experiments:
>
> Unfortunately, here too there seem to be misunderstandings.
> First, the misinterpretation of our theoretical findings as (a)+(b) instead of (i)+(ii) has led to criticism on our experimental protocol. Reviewer's suggestions to focus on a single layer l=log(D), and to make sure that deeper layers should be wider, do not go along with our findings. More generally, any attempt to clearly isolate a single most important layer from others is doomed to fail, as in a real-world dataset (even as simple as MNIST) image features do not have a fixed size (as we emphasize at the end of sec 5). On real-world tasks our findings imply that the width of deep layers is important for modeling correlations across distances, whereas the width of early layers is important for short range correlations. We accordingly compare two simple network architectures:
> - A1: width increases with depth
> - A2: width decreases with depth
> on two prediction tasks:
> - T1: classifying large objects
> - T2: classifying small objects
> Our experiments clearly show that A1 is superior for T1, and vice versa, in compliance with our conclusion. We acknowledge reviewer's claims that much remains to be proven empirically in order to ensure that this conclusion applies in state of the art settings. This is a direction we are currently pursuing in a follow-up work.
>
> An additional misunderstanding arising from the review relates to the performance on MNIST.
> To make sure that we do not introduce bias in terms of objects' locations in an image, our classification tasks include (resized) MNIST digits *located randomly inside an image* (see sec 6).  This is significantly more difficult than the original MNIST benchmark, thus our results are not comparable to those acheived on the latter.
>
> A minor addition to the experiments section has been made for emphasizing the above.
>
> Finally, reviewer writes: "The theory is based on the analysis of ConvAC, and all of a sudden the experiments are conducted on the traditional convolution. This is not rigorous and not professional"
> One of the main roles of our experiments, stated clearly at the opening of sec 6, was to show that our conclusions extend to popular ConvNet architectures - with ReLU activations and max pooling. We do not adapt our analysis to this architecture as in [1], thus a-priori it is unclear to what extent our results capture ReLU and max pooling. This question was addressed through the experiments. We note that the empirical trends we observed with ConvACs were precisely the same - a footnote indicating this was added to the paper.
>
>
> - Related work:
>
> Our focus in this work is on the design of network architectures based on theoretical analyses. We do not compare ourselves to empirical methods for architectural design. The paper should be understood as a contribution to the theoretical understanding of deep learning (via connection to quantum physics), with an empirically demonstrated practical conclusion.
>
> [1]  Cohen and Shashua, ICML 2016.

---

> > ### Comment · AnonReviewer6 · 2018-01-13
> > **Feedback after reading the reviews**
> >
> > Dear authors,
> >
> > Thanks for your feedback. I have gone through the discussion over again. The two implications of Theorem 1 is misunderstood (and therefore the suggested experiment); thanks for your clarification. The explanations are clear to me now and the additional experiments make the paper more solid.
> >
> > I have modified the rating from rejection to weak accept. But, I still have two comments on the experiments.
> >
> > (a) settings (randomly put original image in a larger image) of experiments on MNIST is different and thus harder than other methods. Why not run some baselines of other methods, using the same setting as you do? In this manner, the readers will get a sense of to what extent the improved method will be.
> >
> > For now, the *only* experiment reported in the paper is via Figure 4. It seems to me a little bit weak.
> >
> > (b) I am still not convinced that why you do the experiments on a regular convnet architectures. Since the main theoretical conclusion part is based on convAC, why not directly conduct on convAC and then have a subsection or paragraph saying that your algorithm also generalizes well - "our conclusions extend to popular ConvNet architectures - with ReLU activations and max pooling.".
> >
> > "We note that the empirical trends we observed with ConvACs were precisely the same - a footnote indicating this was added to the paper." what do you mean by "precisely" the same? The same numeric number(error/accuracy)? The same trend or others?
> >
> > I understand the authors must have put a lot of effort into this work and I don't want to kill a theretical-insightful paper. After reading through feedback and revised manuscript, I think it is marginally above the average of ICLR papers, based on my partial and biased experience. But still i would suggest authors to strengthen the experiment part.

---

> > > ### Author Response · Authors · 2018-01-14
> > > **Authors' response to reviewer 6**
> > >
> > > We thank reviewer for the time and effort invested, and for the consideration of our response.
> > >
> > > By "empirical trends we observed with ConvACs were precisely the same", we mean that with ConvACs, as with ReLU ConvNets, "wide-base" architecture had a clear advantage over "wide-tip" in the local task, whereas in the global task it is wide-tip that was clearly superior.  The differences in accuracies between the two architectures in the case of ConvACs was similar to that reported in fig. 4 - on the order of 5% in favor of wide-base/tip on local/global task (respectively).  The overall accuracies with ConvACs were sightly lower than with ReLU ConvNets - a 1% difference on average.  We will add these details to the final version of the manuscript.
> > >
> > > In terms of further experiments, we are currently pursuing an empirical follow-up work in which we evaluate our findings on more challenging benchmarks, with more elaborate architectures.  We hope to release this work to arXiv in the near future and add a reference to it in this manuscript before its finalization.

---

### Decision · Program_Chairs · 2018-01-29
**ICLR 2018 Conference Acceptance Decision**

**Decision:**

Accept (Poster)

**Comment:**

This paper seemingly joins a cohort of ICLR submissions which attempt to port mature concepts from physics to machine learning, make a complex and non-trivial theoretical contribution, and fall short on the empirical front. The one aspect that sets this apart from its peers is that the reviewers agree that the theoretical contribution of this work is clear, interesting, and highly non-trivial. While the experiment sections (MNIST!) is indubitably weak, when treating this as a primarily theoretical contribution, the reviewers (in particular 6 and 3) are happy to suggest that the paper is worth reading. Taking this into account, and discounting somewhat the short (and, by their own admission, uncertain) assessment of reviewer 5, I am leaning  towards pushing for the acceptance of this paper. At very least, it would be a shame not to accept it to the workshop track, as this is by far the strongest paper of this type submitted to this conference.